# Pseudo-topotactic conversion of carbon nanotubes to T-carbon nanowires under picosecond laser irradiation in methanol

Jinying Zhang [1], Rui Wang[1], Xi Zhu[2], Aifei Pan[3], Chenxiao Han[1], Xin Li[1], Dan Zhao[1], Chuansheng Ma[4], Wenjun Wang[3], Haibin Su[2] & Chunming Niu[1]

Pseudo-topotactic conversion of carbon nanotubes into one-dimensional carbon nanowires is a challenging but feasible path to obtain desired diameters and morphologies. Here, a previously predicted but experimentally unobserved carbon allotrope, T-carbon, has been produced from pseudo-topotactic conversion of a multi-walled carbon nanotube suspension in methanol by picosecond pulsed-laser irradiation. The as-grown T-carbon nanowires have the same diameter distribution as pristine carbon nanotubes, and have been characterized by high-resolution transmission electron microscopy, fast Fourier transform, electron energy loss, ultraviolet–visible, and photoluminescence spectroscopies to possess a diamond-like lattice, where each carbon is replaced by a carbon tetrahedron, and a lattice constant of 7.80 Å. The change in entropy from carbon nanotubes to T-carbon reveals the phase transformation to be first order in nature. The computed electronic band structures and projected density of states are in good agreement with the optical absorption and photoluminescence spectra of the T-carbon nanowires.

---

[1] Center of Nanomaterials for Renewable Energy, State Key Laboratory of Electrical Insulation and Power Equipment, School of Electrical Engineering, Xi'an Jiaotong University, Xi'an 710054, China. [2] Divisions of Materials Science, Institute of Advanced Studies, Nanyang Technological University, 50 Nanyang Avenue, Singapore 639798, Singapore. [3] State Key Laboratory for Manufacturing System Engineering, Xi'an Jiaotong University, Xi'an 710054, China. [4] The School of Electronic and Information Engineering and State Key Laboratory for Mechanical Behaviour of Materials, Xi'an Jiaotong University, Xi'an 710049, China. Correspondence and requests for materials should be addressed to J.Z. (email: jinying.zhang@mail.xjtu.edu.cn) or to H.S. (email: hbsu@ntu.edu.sg)

From naturally existing soft graphite and super-hard diamond to synthetic ballistic conducting nanotubes and semiconducting graphene, carbon chemistry is rich and varied. Carbon nanotubes (CNTs), one-dimensional (1D) metastable carbon nanostructures, have attracted significant attention[1–6] in recent decades due to their unique physical and chemical properties, despite not having accurate positions in the temperature–pressure (T-P) phase diagram of carbon[7]. Template growth of 1D carbon nanostructures (CNTs[8, 9], carbon linear chain[10, 11], diamond nanowires (NWs)[12, 13], and graphene nanoribbons[5, 6]) in the inner chambers of CNTs has been well studied. However, the pseudo-topotactic conversion of CNTs into carbon NWs has not yet been reported, to the best of our knowledge. Nevertheless, the investigation of metastable carbon allotropes in the T–P phase diagram between graphite and cubic diamond phases has been rewarding both experimentally and theoretically, and remains a great challenge. Lonsdaleite (hexagonal diamond), discovered in 1967 associated with the production of diamond[14], was found to have a hexagonal lattice structure (P6$_3$/mmc). Linear carbyne was produced by shock compression in 1991[15]. The structure of the well-used glassy carbon and chaoite is still under debate. A body-centered cubic structure with eight atoms in the unit cell (C$_8$, Im $\overline{3}$[16], actually Im $\overline{3}$m[17]) has been produced by the condensation of a carbon plasma stream[16]. The C$_8$ like nanoparticles were also claimed to be synthesized from amorphous carbon in aqueous solution by nanosecond pulsed-laser irradiation[18, 19]. BC$_8$ (Ia $\overline{3}$)[20, 21], bct C$_4$ (I4/mmm)[22], M-carbon (C2/m)[23], prismane C$_8$[24], and T-carbon (Fd $\overline{3}$m)[25] carbon allotropes have also been theoretically predicted. The X-ray inelastic scattering spectra of graphite under high pressure[26] were claimed to partially include bct C4[22] and M-carbon[23] with negligible peaks.

The synthesis of metastable carbon structures became feasible with the emergence of new synthetic technology. Extremely high temperatures and pressures can be reached and subsequently quenched to synthesize and trap metastable carbon allotropes by the interactions between pulsed laser and carbon materials. However, carbon particles were usually produced by laser irradiation, high pressure compression, or plasma chemical vapor deposition (CVD). Diamond particles[27] and C$_8$ like metastable carbon particles[18, 19] have been produced from amorphous carbon by nanosecond pulsed lasers. Diamond nanoparticles and carbon-nano-onions have also been found to be reversibly transformed in alcohol solution by nanosecond laser irradiation[28]. Diamond nanoparticles instead of diamond NWs have also been produced from CNTs by laser irradiation[29]. Different structural forms of carbon were also produced by laser irradiation of CNT films under a nitrogen atmosphere[30]. The synthesis of carbon NWs, especially shape controlled NWs with sp$^3$ hybridization, is still a great challenge.

Here, we describe the synthesis and characterization of a previously unobserved carbon allotrope (T-carbon) NW by pseudo-topotactic conversion of multi-walled CNTs (MWCNTs) in methanol solvent by picosecond laser irradiation under a nitrogen atmosphere.

## Results

**Preparation of T-carbon NWs.** MWCNTs were prepared by a CVD method to have diameters around 10–20 nm[31, 32] and lengths of dozens of micrometers, and subsequently shortened by a sonication method[33] to improve their dispersion. The high-resolution transmission electron microscopy (HRTEM) image of the shorted MWCNTs is shown in Fig. 1a. About 24% of the MWCNTs were found to be ~100–200 nm after the shortening process (Supplementary Fig. 1). The shortened MWCNTs (0.5 mg) were well dispersed in absolute methanol (40 ml). The suspension was then transferred into a self-designed

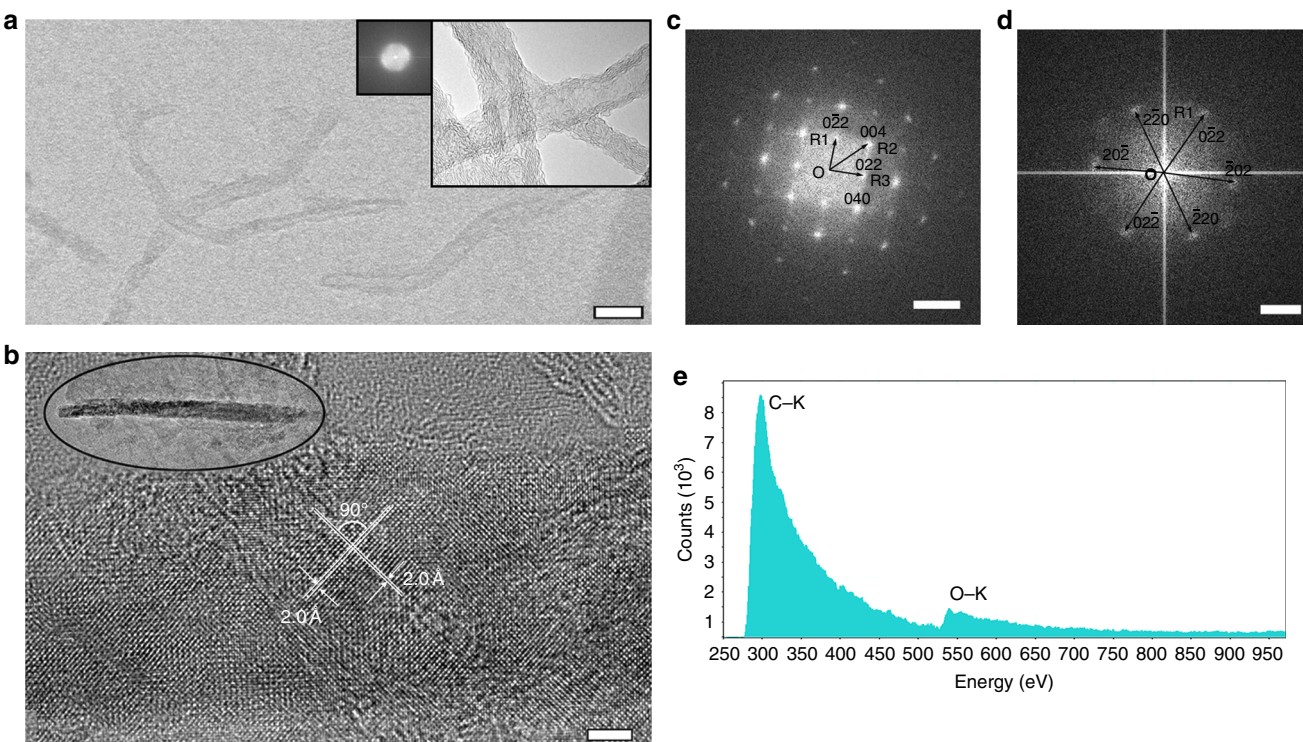

**Fig. 1** Structure transformation of T-carbon nanowires from MWCNTs. HRTEM images of **a** shortened MWCNTs (insets show the structure and FFT pattern of pristine MWCNTs) (scale bar = 20 nm) and **b** carbon nanowire produced from laser irradiation (inset: low magnification, scale bar = 5 nm). FFT patterns of **c** the nanowire corresponding to **b** (scale bar = 5 nm$^{-1}$), and **d** another nanowire (scale bar = 2 nm$^{-1}$). **e** EELS spectrum corresponding to **b** after subtraction of carbon film background

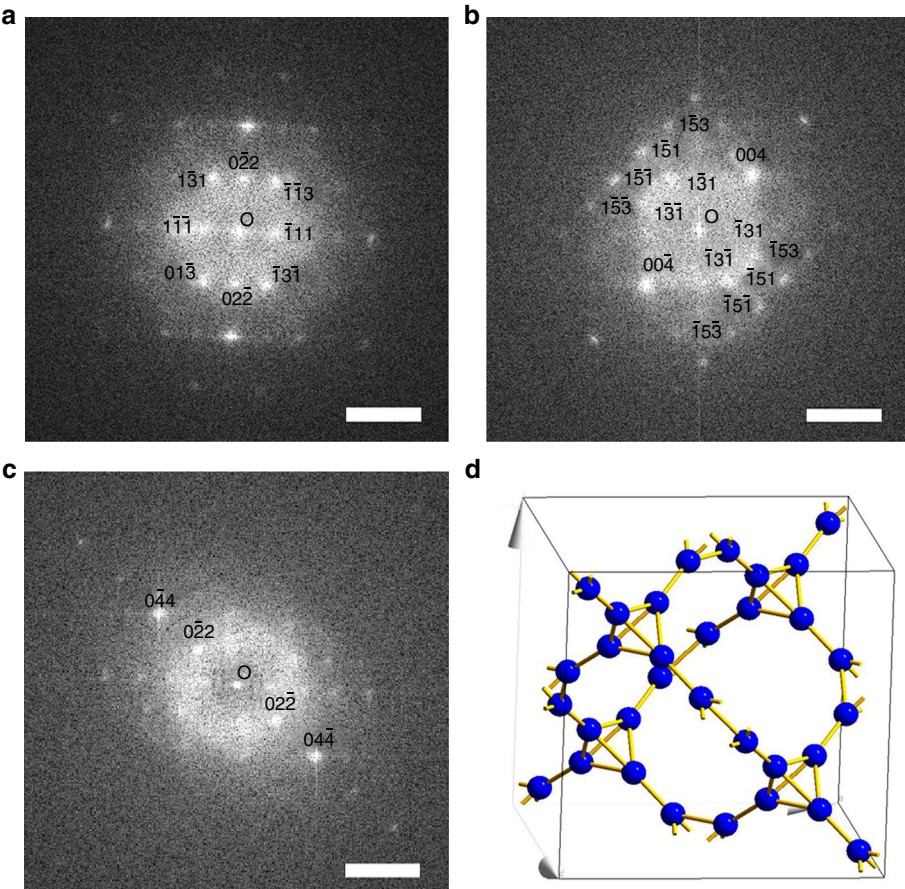

**Fig. 2** FFT patterns and structural model of a T-carbon nanowire. FFT patterns with **a** zone axis of [2, 1, 1], **b** zone axis between [3, 1, 0] and [5, 1, 0] with an angle from **a** of 27°, and **c** zone axis close to [5, 1, 1] with an angle from **a** of 18° and an angle from **b** of 10° (*scale bar* = 5 nm$^{-1}$). **d** Structural model of T-carbon (Fd$\bar{3}$m, lattice constant = 7.80 Å)

quartz container with an optical path length of 40 mm and preserved under a nitrogen atmosphere. A Q-switched laser with a wavelength of 532 nm (doubled from a Nd:VAN laser with a wavelength of 1064 nm), pulse duration of 10 ps, repetition frequency of 1000 Hz, and pulse power of 75 mW was applied. The laser was focused 0.5 cm away from the front wall of the self-designed container with a beam size of 0.5 mm. The suspension was irradiated by the laser beam for 1 h while it was kept under a nitrogen atmosphere and stirred with a magnetic stirring bar. The suspension became transparent after the laser reaction. The as-produced gas was collected for further analysis.

**Lattice structures of T-carbon.** Many unreacted MWCNTs and amorphous phases in addition to the NWs were observed by HRTEM from the suspension after laser irradiation (Supplementary Fig. 2). The NWs were observed to have the same crystal planes and interplanar crystal spacings, which are distinguishable from unreacted MWCNTs (Fig. 1a and Supplementary Fig. 3a) and amorphous phases (Supplementary Fig. 3b). Two sets of crystal planes with an interplanar crystal spacing of 1.95 ± 0.01 Å intersected with an interplanar angle of 90 ± 2° were observed (Fig. 1b). A fast Fourier transform (FFT) pattern calculated from the HRTEM image is shown in Fig. 1c. The FFT pattern shows a square pattern with vector OR$_1$ corresponding to a lattice fringe spacing of 2.76 ± 0.01 Å and vector OR$_2$ corresponding to a lattice fringe spacing of 1.95 ± 0.01 Å, indicating a cubic or tetragonal crystal system. FFT patterns were used instead of selected area electron diffraction (SAED) for structure analysis since the NWs were

unstable under SAED measurement conditions where an electron flux of several orders of magnitude larger than $10^5$ [e] nm$^{-2}$ s$^{-1}$ was used. Tilting the NWs to other desired zone axes is a big challenge since the NWs are only about 10–20 nm wide (hard to observe Kikuchi lines). The analysis of many NWs with different zone axes parallel to the electron beam direction was adopted for further lattice characterization. The FFT of another NW gives a hexagonal pattern corresponding to lattice fringe spacing of 2.76 ± 0.01 Å as shown in Fig. 1d. The square and hexagonal FFT patterns of the as-produced NWs suggest a cubic crystal lattice.

The reagents in the reaction system include MWCNTs and methanol only. Hence, only carbon, oxygen, and hydrogen elements are contained in the products. Electron energy loss spectroscopy (EELS) was adopted to determine the valence, chemical bonding, and atomic composition of the as-produced NWs (Fig. 1e)[34]. Background correction was carried out for the EELS spectra before calculation of the $\sigma/\pi$ ratio to eliminate the effects of carbon film. Only the electron transition from 1-$s$ level to $\sigma^\star$ band was observed from the carbon $k$-edge spectrum, where the electronic transition from 1-$s$ level to $\pi^\star$ band is negligible (Fig. 1e). The $(\pi^\star/(\pi^\star + \sigma^\star))$ ratio of the NWs was estimated to be 0.04 ± 0.01. The $sp^2$ ratio was calculated to be 16 ± 4% due to the surrounding amorphous structures. The carbon NWs are supposed to be composed of only $sp^3$ carbon, which is similar to diamond[35]. The $(\pi^\star/(\pi^\star + \sigma^\star))$ ratio of the unreacted MWCNTs (Supplementary Fig. 3a) was estimated to be 0.13 ± 0.01 according to the carbon $k$-edge (Supplementary Fig. 3c). The ratio of $sp^2$ carbon is calculated to be 51.6 ± 0.03%[13, 36]. The high $sp^3$ ratio of the NWs is consistent with the T-carbon structure.

A small oxygen $k$-edge transition ($25 \pm 2\%$) in addition to carbon $k$-edge was also observed from the EELS spectrum of NWs. No oxygen was detected in the MWCNT spectrum, while strong oxygen $k$-edge ($86 \pm 3\%$, Supplementary Fig. 3d) was obtained from the amorphous structure after laser irradiation. The NWs are all surrounded by amorphous structures, which were likely produced from methanol during laser irradiation. Covalent bonds might be produced between the amorphous structures and NWs to passivate the T-carbon NWs during laser irradiation. The small oxygen $k$-edge can be attributed to the surrounding amorphous structures produced from laser interaction with methanol because methanol is the only oxygen source in the reaction system. To further confirm that the oxygen signal is related to the methanol conversion, methanol solvent has been irradiated under the same conditions. Large amounts of the same amorphous structures were produced. It is clear that EELS analysis has shown that $sp^2$ hybridization in MWCNTs has been transformed to $sp^3$ hybridization. And furthermore, the amorphous coating on the surface of NWs contained oxygen arising from methanol conversion.

The NWs produced from laser irradiation of MWCNTs in methanol under nitrogen atmosphere have been so far confirmed to be $sp^3$ hybridized carbon NWs with a cubic crystal lattice. The FFT patterns of one single carbon NW at different tilting angles (Fig. 2) were used to further confirm the crystal lattice of the carbon NWs to be T-carbon (Fd $\overline{3}$m (227)) with a lattice constant of 7.80 Å. The structure is diamond-like with each carbon atom replaced by a carbon tetrahedron (Fig. 2d), consistent with the theoretically predicted T-carbon with a lattice constant of 7.52 Å[25]. The carbon atoms of T-carbon occupy the Wyckoff position 32e ($x$, $x$, $x$) with $x = 0.0706$. There are two distinct C–C bond lengths of 1.56 Å (intra-tetrahedron) and 1.47 Å (inter-tetrahedron). Although, similar to the diamond structure, each carbon in T-carbon is covalently bonded with four adjacent carbon atoms via $sp^3$ hybridization, localization in a group of four carbon atoms takes place that leads to substantial deviation of C–C bonding from the ideal $sp^3$ bonding in the diamond structure. This deviation results in an appreciable electron transfer from intra-tetrahedron bonds to inter-tetrahedron ones. Therefore, the electron density of inter-tetrahedron bonds is much higher than that of intra-tetrahedron bonds, which leads to shorter bonds of the inter-tetrahedrons compared with the intra-tetrahedrons. The inter-tetrahedron bonds are much stronger than the intra-tetrahedron ones, a distinctive structural feature in comparison to the diamond structure.

The FFT patterns shown in Fig. 1c, d are consistent with the [1, 0, 0] and [1, 1, 1] diffraction patterns of T-carbon, respectively. The corresponding crystal planes are indicated in the FFT patterns. The two sets of crystal planes shown in Fig. 1b correspond to the (0, 0, 4) and (0, 4, 0) crystal planes, respectively. The FFT pattern shown in Fig. 2a is consistent with the [2, 1, 1] diffraction pattern of T-carbon for both interplanar distances and angles. A new FFT pattern (Fig. 2b) was obtained after tilting 27° from Fig. 2a with the electron beam direction between [3, 1, 0] and [5, 1, 0] of the T-carbon. The NW was further tilted to have an angle relative to Fig. 2a of 18° and to Fig. 2b of 10°, resulting in the FFT pattern shown in Fig. 2c. The electron beam direction of Fig. 2c was calculated to be close to [5, 1, 1]. The interplanar distances are consistent with each other in all FFT patterns, which are listed in Supplementary Table 1. The interplanar angles measured in the FFT patterns are also consistent with the calculated values. The comparison data are listed in Supplementary Table 2. Only lattice planes with either all even, whose sum is a multiple of 4, or all odd indices were observed in the FFT patterns due to the extinction rules for T-carbon, which are demonstrated in the Supplementary Methods.

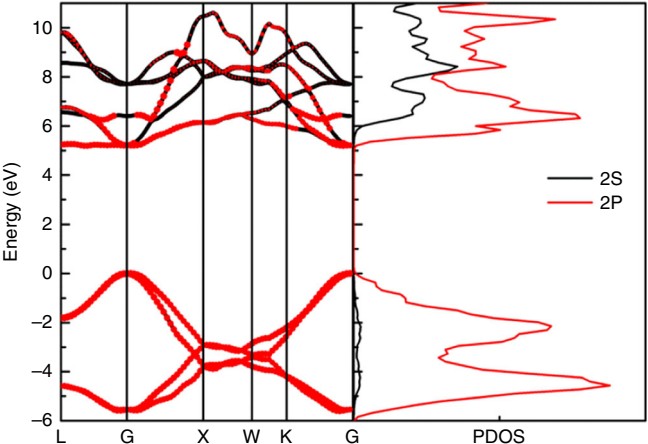

**Fig. 3** $G_0W_0$ corrected band structure and projected density of states of T-carbon

**Electronic band structures of T-carbon.** The electronic band structure was investigated using density functional theory (DFT)[37]-based Vienna ab initio simulation package[38, 39] by applying the projector augmented wave[40] potential with Perdew–Burke–Ernzerhof[41] function. The subsequent $G_0W_0$ corrected band structure with direct band gap of 5.10 eV at the G point and projected density of states (PDOS) are presented in Fig. 3. The valence band maxima and conduction band minima exhibit $p$-orbital character. The electronic band structures and PDOS of T-carbon provide a basis to corroborate the ultraviolet–visible (UV–Vis) absorption and photoluminescence (PL) of the products (Fig. 4). The solution after laser irradiation of MWCNT suspension in methanol was directly used for UV–Vis and PL measurements, while the methanol after laser irradiation with the same conditions and the suspension of MWCNTs in methanol before laser irradiation were used for comparison. Several absorption peaks were detected from the T-carbon NWs in the range 220–320 nm (Fig. 4a, *black*). The three main peaks are at 225 nm (5.51 eV), 250 nm (4.96 eV), and 276 nm (4.49 eV). Contrary to the spectrum of T-carbon, there are no peaks at 225, 250, and 276 nm presented in the spectra measured from either the methanol solvent after laser irradiation (Fig. 4a, *blue*) or the MWCNT suspension before laser irradiation (Fig. 4a, *red*). The full scale absorption spectra are shown in Supplementary Fig. 4. Strong PL at 436 nm (2.84 eV) with a quantum yield of $5.4 \pm 0.2\%$ was observed from the T-carbon NW products (Fig. 4b, *black* and Supplementary Fig. 5) with excitation at 372 nm (3.33 eV). Similar to the observation for absorption peaks, the PL emission was neither observed from the methanol solvent after laser irradiation (Fig. 4b, *blue*) nor from the MWCNT suspension (Fig. 4b, *red*). The shape of the absorption spectrum corresponds with the DOS of the conduction bands, while the emission is related to the DOS of the valence bands. Moreover, the temperature-dependent PL data are presented in Supplementary Fig. 6. The PL intensity increases with decreasing temperature, which clearly suggests that T-carbon has a direct band gap. With the computed gap (5.10 eV) and the first absorption peak 4.54 eV (273 nm) as shown in the *inset* of Fig. 4a, the exciton binding energy is 560 meV, which is eight times the exciton binding affinity of diamond (70 meV)[42]. The strong exciton binding energy enables the observation of exciton features in optical measurements at room temperature. The *inset* in Fig. 4a shows the computed absorption spectrum with pronounced peaks at 5.48 eV (226 nm), 5.17 eV (240 nm), and 4.54 eV (273 nm) in good agreement with the experimentally measured absorption spectrum. Moreover, the shape of the

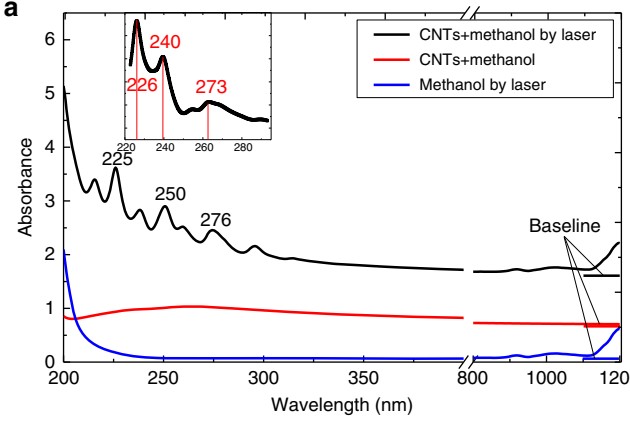

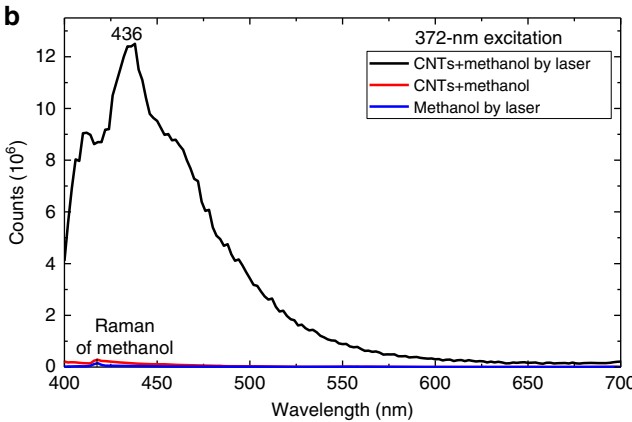

**Fig. 4** Optical absorption and photoluminescence spectra. Spectra of crude T-carbon products (*black*) compared to pristine carbon nanotube suspension (*red*) and methanol irradiation products (*blue*). **a** UV–Vis absorption (absorption lines were vertically shifted) (*inset* shows the simulated absorption curve). **b** Photoluminescence spectra

absorption spectrum of T-carbon shares more features with that of diamond, with respect to those of graphene nanoribbons and single-walled CNTs as shown in Supplementary Fig. 7. The first transition peak of T-carbon (4.6 eV) is also closer to that of diamond (7.0 eV) than that of graphene nanoribbons and single-walled CNTs (1.5 eV), further confirming that T-carbon is composed of $sp^3$ rather than $sp^2$ carbon. The Raman spectrum of the products after laser irradiation was obtained by using an excitation laser of 514 nm (Supplementary Fig. 8, *black*), and is distinguishable from that of diamond structures (Supplementary Fig. 8, *red*).

**Size distribution of T-carbon NWs**. The pseudo-topotactic conversion of T-carbon NWs from MWCNTs in methanol during picosecond laser irradiation was further confirmed by their diameter distributions. The lengths and diameters of ~100 T-carbon NWs were measured from large area HRTEM images. Most of the T-carbon NWs are in the range 100–150 nm, which was not affected by laser irradiation power between 75–95 mW (Fig. 5a). The percentage of NWs with a length of >200 nm is relatively small, likely due to the limitation of laser-CNT interaction areas. The diameters of the T-carbon NWs are mainly distributed between 10–20 nm, with an average diameter of 11.8 ± 2.8 nm (Fig. 5b). The diameter distribution of the T-carbon NWs (Fig. 5b) is consistent with that of shortened MWCNTs (11.7 ± 2.2 nm) used for laser irradiation, further confirming the successful pseudo-topotactic conversion of MWCNTs into T-carbon NWs.

## Discussion

The phonon density of states, specific heat, and entropy of graphite, diamond, CNT, and T-carbon were calculated with the quasi-harmonic approximation as shown in Supplementary Fig. 9 and Supplementary Table 3[43]. The computed longitudinal optical (LO) mode of diamond (CNT (6,6)) is 35 (45) THz, which agrees nicely with the experimentally measured value of 36 (44) THz by inelastic X-ray scattering[44–46]. The computed LO mode of T-carbon is 53 THz (1767 cm$^{-1}$), in good agreement with the reported value, 1760 cm$^{-1}$[25]. The LO mode of T-carbon has a higher frequency than that of diamond due to the strain stored in the tetrahedrons. A distinct variation of entropy from the MWCNTs to T-carbon was involved in the pseudo-topotactic transformation of MWCNTs to T-carbon NWs, indicating this transformation to be a first-order phase transformation.

It is well known that pulsed-laser-induced liquid–solid interfacial reactions are a fast and far-from-equilibrium process. Instantaneous high temperatures and pressures are produced in the laser–materials interface and then immediately quenched. The metastable carbon allotropes can be formed and preserved as final products. The overall shape of the CNT auto templates was preserved to yield NW products by irradiation with a picosecond pulsed laser.

The transformation mechanism of T-carbon NWs from MWCNTs remains to be explored. Narayan and Bhaumik[47] observed the conversion of nanodiamond from amorphous carbon film deposited on sapphire by irradiation with a nanosecond laser. The formation of nanodiamonds by a melting mechanism via a "super undercooled" state was proposed. Instead of a solid thin film of carbon, our starting materials were individualized MWCNTs suspended in methanol, and a picosecond laser was used. Rather than an asymmetric quenching environment of thin film transition (air on the surface, carbon in the lateral direction, and sapphire underneath), we have a symmetric environment of individual MWCNTs surrounded by liquid methanol. The energy absorption and dissipation are confined to the vicinity of the MWCNTs at a much smaller scale (15 nm (D) × 200 nm (L)). Furthermore, we have observed gas evolution and amorphous carbon formation from methanol decomposition. Given the highly kinetically controlled nature of the reaction (sensitive to any condition changed), the setup in this work leads to a different mechanism of structural transition leading to the formation and stabilization of T-carbon NWs. The difference lies in the time scales of energy transfer from the laser to the MWCNTs and their subsequent ultrafast quenching. We believe these are key factors contributing to the formation and stabilization of T-carbon NWs from MWCNTs. The shapes of the MWCNTs were retained due to the short time scales and reduced heating effect of the picosecond laser. Hydrogen gas generation during femtosecond laser irradiation as a function of the standard enthalpy change of various alcohol solutions[48] demonstrated that methanol has a profound inhibitory effect on oxidation. Hydrogen gas, carbon monoxide, carbon dioxide (Fig. 6), and an amorphous-like carbon (Supplementary Fig. 3b) were produced by the laser dissociation of methanol molecules. The detection of hydrogen gas from the reaction further confirmed the decomposition of methanol during laser irradiation since the only possible source of hydrogen is the methanol solvent. The dissociated hydrogen might play a role in surface $sp^3$ dangling bond passivation. The amorphous-like carbons further adhere to the surface of T-carbon NWs (Fig. 1b) to act as a passivation layer. The preservation of the shape of MWCNTs to T-carbon NWs might be due to the higher absorption of 532 nm photons by MWCNTs than that of NWs, where the absorption for MWCNT suspension is more than twice that of NW or methanol solvent at 532 nm (Supplementary Fig. 4). The MWCNTs were multiphoton-excited to a high energy

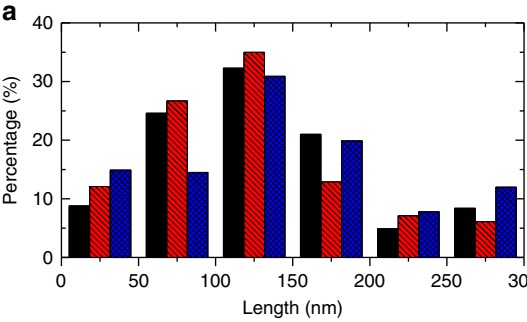
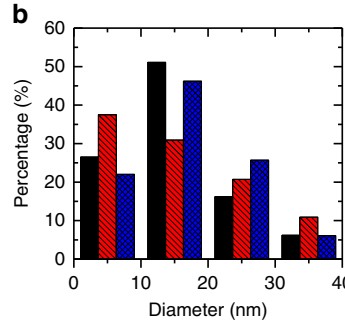

**Fig. 5** The length and diameter distributions of T-carbon nanowires produced using different laser powers. **a** Length distribution. **b** Diameter distribution. *Black* = 75 mW, *red* = 85 mW, *blue* = 95 mW

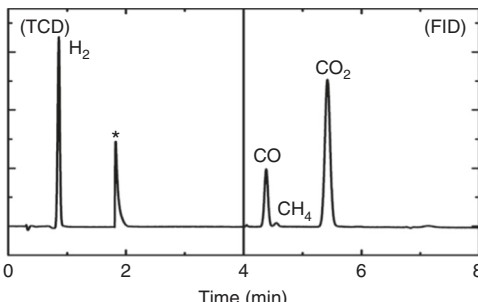

**Fig. 6** The gas chromatogram of the gas from laser irradiation of MWCNT suspension in methanol (* = from equipment disturbance caused by pressure fluctuation due to switching of injection valves). TCD, thermal conductivity detector; FID, flame ionization detector

and transformed to T-carbon NWs and further trapped at room temperature.

In summary, we have demonstrated the successful synthesis of a previously predicted but experimentally unobserved carbon allotrope, T-carbon, by picosecond pulsed-laser irradiation of a MWCNT suspension in methanol. The carbon allotrope was characterized by HRTEM, FFT, EELS, UV–Vis, PL and identified as T-carbon, with a NW morphology inherited from the starting MWCNTs via pseudo-topotactic conversion. The NWs have a diamond-like lattice where each carbon atom is replaced by a carbon tetrahedron (Fd $\overline{3}$m (227)). The lattice constant determined from FFT patterns is 7.80 Å, which is consistent with the theoretical prediction of 7.52 Å. The diameters of most T-carbon NWs are in the range 10–20 nm, similar to the diameter distribution of the starting shortened MWCNTs. The T-carbon structure exhibits a direct band gap of 5.10 eV. The computed electronic band structures and PDOS are in good agreement with the UV–Vis absorption and PL emission of the T-carbon NWs. The phase transformation from MWCNT to T-carbon is first order, as shown by the change in entropy of these two phases. Ultra-short pulsed-laser irradiation is an effective path to the synthesis of metastable carbon structures, where the methanol solvent plays a critical role in the synthesis and preservation of T-carbon NWs.

## Methods

**Experiment**. The MWCNTs were synthesized by a catalytic CVD method using Co/Fe-Al$_2$O$_3$ as catalyst, ethylene as carbon source, and further shortened by a ultra-sonification method[33]. The shortened MWCNTs (0.5 mg) were well dispersed in 40 ml absolute methanol (99.9%, ThermoFisher) before laser experiment. The suspension was preserved in a self-designed quartz container with an optical path length of 50 mm under an atmosphere of nitrogen. A Q-switched laser with a wavelength of 532 nm (doubled from a Nd:VAN laser with a wavelength of 1064 nm), pulse duration of 10 ps, repeating frequency of 1000 Hz, and laser pulse power of 0.075 mJ was focused on the front side of the bottle (0.5 cm away from the front

wall) with 0.5 mm beam size. During the laser irradiation, the solution was kept stirring with a magnetic stirrer. HRTEM images, EELS spectra, and FFT patterns were acquired using an FEI Titan G$^2$ 60-300 transition electron microscope equipped with a field-emission gun (acceleration voltage: 300 kV). Optical absorption and PL spectra were taken from UV–Vis–NIR spectroscopy (V670, Jasco) and a steady state FLS980 PL spectroscopy (Edinburgh Instruments), respectively. Raman spectroscopy was taken in a back-scattering geometry using a single monochromator with a microscope (Reinishaw inVia) equipped with CCD array detector (1024 × 256 pixels, cooled to −70 °C) and an edge filter. The samples were excited by 514.5 nm Argon ion laser. The gas mixture from the laser irradiation was characterized by gas chromatography (GC-456, Bruker) equipped with thermal conductivity detectors and flame ionization detectors. Samples were protected under a nitrogen atmosphere and transferred inside a glove box to prevent contamination.

**Calculation**. A 500 eV kinetic energy cutoff and 8 × 8 × 8 *k*-points sampling are used for the ground state calculation. The geometry structures are fully optimized till the atomic forces are converged to $1 \times 10^{-5}$ eV Å$^{-1}$. The G$_0$W$_0$ approximation was used for the quasi-particle corrections to the DFT band gaps. The plasmon-pole approximation[49] was used to treat the screening effect. The electron–hole interactions were resolved by the Bethe–Salpeter equation (BSE)[50] approaches as:

$$(E_{ck} - E_{vk})A^S_{vck} + \sum_{k'v'c'} \langle vck|K^{eh}|v'c'k'\rangle A^S_{vck} = \Omega^S A^S_{vck} \qquad (1)$$

where $A^{mS}_{vck}$ is the exciton's amplitude, $K^{eh}$ is the kernel of electron–hole interaction, $|ck\rangle$ and $|vk\rangle$ are the quasi-particle states (electron and hole), $\Omega^S$ is the excitation energy, and $E_{ck}$ and $E_{vk}$ are the quasi-particle energies. To avoid problematic image effects from nearby cells, we apply a box-shape truncated super cell for dealing with the Coulomb interaction. The phonon, entropy, and specific heat are computed with the quasi-harmonic approximation[43].

**Data availability**. The authors declare that the data supporting the findings of this study are available within the article and Supplementary Information files, or from the corresponding authors upon reasonable request.

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

## Acknowledgements

This research is supported by the National Natural Science Foundation of China (no. 51302210) and the Fundamental Research Funds for the Central Universities. The TEM work was done at the International Center for Dielectric Research. Work at NTU (Singapore) was supported in part by MOE Tier—two grants (nos. MOE2013-T2-2-049 and MOE2013-T2-2-002) and the Society of Interdisciplinary Research. J.Z. is supported by the Cyrus Tang Foundation through Tang Scholar Program.

## Author contributions

J.Z. designed the research and supervised the experiments, data analysis, and paper writing. R.W. performed the experiments. A.P. operated the laser equipment. W.W. supervised the laser operation. X.Z. performed the calculations. H.S. supervised the calculations. C.H. assisted the sample preparation and experiments. C.M. performed the HRTEM measurements. X.L. and D.Z. assisted the TEM observation. H.S. and C.N. assisted the paper writing. All authors have participated in the discussions.

## Additional information

**Competing interests:** The authors declare no competing financial interests.

