## [Peer Review File · Nature Communications]

Reviewers' comments:

Reviewer #1 (Remarks to the Author):

A new carbon allotrope, i.e., T-carbon, with the structure formed by each carbon in cubic diamond replaced with a carbon tetrahedron, which was predicted by first-principles calculations several years ago, is now asserted to be obtained in the form of nanowires from carbon nanotubes in methanol under picosecond pulsed-laser ablation. The synthesis procedure is described, and the measurements and analyses of HRTEM, FFT, EELS, UV-Vis, and PL are presented, which appear to show, compatibly, the diamond-like structure "is consistent with a theoretically predicted T-carbon structure (lattice constant 7.52 Å)". If the evidence is solid, this work would be a breakthrough in the field of carbon science, as it is the third identified three-dimensional periodic structure of carbon allotropes (the other two include cubic diamond and lonsdaleite). It will also deepen our further understanding on the diversity of carbon bonding. I would recommend publication of this manuscript provided that the following points are carefully clarified:

1. I note that the authors use HRTEM and FFT to characterize the new structure. They only provide high-index FFT patterns. Can the low-index FFT pattern be given? Or can a direct electron diffraction image (other than FFT pattern) with low index on the new structure be provided? Such information will further convince the reader and strengthen the statement.
2. The shape of EELS for carbon (Fig. 1e) looks a bit strange. No low energy loss peak is observed, which may probably lead to a misjudgment on the ratio of sigma/pi orbitals.
3. In the whole manuscript, "tetra-diamond" should be changed to "T-carbon", as the latter is the widely accepted name for such a structure of carbon.
4. Typos. In title: "Psuedo-topotactic" should be "Pseudo-topotactic"; pages 1 & 4, several places, "tpotactic" should be "topotactic"; there are other typos throughout the text.

Reviewer #2 (Remarks to the Author):

The authors report conversion of carbon nanotubes into tetra diamond nanowires by picosecond laser irradiation. Nanotubes are suspended in methanol and the conversion into tetra diamond is claimed by some pseudo-topotactic growth, because size distribution of nanotubes and tetra diamond nanorods remains the same. There are so many technical loose ends which needs to be addressed and tied down firmly, before this work can be considered for publication.

(1) This conversion from nanotubes to tetra diamond involves a distinct change in structure (entropy), which implies it is a first-order phase transformation.

(2) What is the template for pseudo-topotactic growth and resulting first-order phase transformation?

(3) Why NW samples were destroyed under Selected-area-electron-diffraction (SAED), which should give more definitive results than FFT patterns on the structural characterization.

In reviewer's opinion, nanotubes are being melted under a super undercooled state and converted into diamond by the mechanism proposed in recent exciting breakthroughs (J. Appl. Phys 118, 215303(2015)).

Reviewer #3 (Remarks to the Author):

The authors claim to have synthesized tetra-diamond carbon nano-wires (NWs) by irradiating multi-walled carbon nanotubes (MWCNTs) with a pico-second laser. The claim represents a novelty regarding the use of pico-second laser and the experimental identification of the tetra-diamond carbon phase. But there are comments from experimental view that need to be addressed:

1. The authors chiefly support the synthesis of the tetra-diamond NW by matching the measured lattice constant of the laser-synthesized material to the theoretical predictions of the reference 25.

For that, the experimental measurement is obtained from Fast Fourier (FFT) generated from high resolution transmission electron microscopy (HREM) images. Nevertheless, neither comparisons between the FFT of the tetra-diamond NW with NWs existing in different phase nor with the starting MWCNT are provided. This does not permit clearly separate the tetra-diamond phase from other products that can also be created by the laser treatment.

2. Electron energy loss spectroscopy (EELS) is used to quantify the sp³ bonds since the reference 25 predicts massive presence of such bonds in the tetra-diamond phase. However, this EELS discussion in the manuscript lies mostly on amorphous and MWCNT forms and it is unclear the EELS information from the laser-created NW.

3. In the reference 25, it is predicted 3.0 eV band gap for the new carbon phase. In the manuscript, it is shown, from UV-Vis absorption, bands corresponding to 5.51, 4.96 and 4.49 eV. Hence, the measurements are inconsistent with the predictions.

4. A minor issue. A Nd:YAG laser has wavelength of 1064 nm that can be doubled in frequency to produce 532 wavelength laser beam. The authors should consider this at the third paragraph in the end of the first page when describing the methods.

The manuscript has some typos and needs to be reviewed with respect to proper use of the writing:

1. In the abstract, the second sentence has the typo 'pseudo-tpotactic'.

2. The term 'pulse width' for the temporal dynamics of the laser pulse is not commonly used. Instead, usually one writes time duration. Consider revising this at the third paragraph in the end of the first page.

3. To a more proper formal language, it should be considered alternative terms for 'lots of' and 'it is hard'. Both terms appear in the fourth paragraph.

4. In the first sentence of the sixth paragraph, 'has' must be changed to have since its subject is NWs in the plural.

5. In the end of the eighth paragraph, change 'The PL intensity increase' to The PL intensity increases since subject is third person.

6. In the end of the tenth paragraph, 'NWs might due to' should be changed to 'NWs might be due to'.

From a broader view, MWCNTs have been submitted to laser irradiation as in del Pino, J. Appl. Phys, 2014. Although a nano-second laser was used in this work, its conclusions indicate the formation of different phases of carbon. Thus, such a work and related ones should be mentioned and cited.

In general, the comments 1-4 pose important questionings on the experimental evidences that state the synthesis of the tetra-diamond NW using the laser method. And, in this stage, these issues represent strong inconsistencies. If the authors can convincingly provide more experimental evidences from the data in the manuscript, maybe, Raman measurements could be performed, and/or clarify those comments, a further review round should be considered.

Reviewer #4 (Remarks to the Author):

The authors report on the pseudo-topotactic conversion of carbon nanotubes to tetra-diamond nanowires in methanol under irradiation of picosecond laser and claim the formation of a new carbon allotrope. However, based on the following considerations, this reviewer does not feel that the aforementioned goal has been achieved in an unequivocal manner and, since reasonable doubts are in the manuscript, I do not consider that this work meets the criteria for publication in Nat. Commun.

Abstract:

This reviewer wonders whether the term "laser ablation" is suitable to describe the preparation of the carbon NWs, since no material is removed from a solid or liquid surface but transformed with a laser beam. Maybe "picosecond pulsed-laser irradiation" is more suitable and accurate.

Concerning the text:

"The as-grown tetra-diamond NWs have the same diameter distribution as pristine MWCNTs and been demonstrated by high resolution transmission electron microscopy, fast Fourier transform, electron energy loss, ultraviolet-visible, and photoluminescence spectroscopies to have a diamond-like lattice with each carbon replaced by a carbon tetrahedron and a lattice constant of 7.80 Å."

Techniques such as UV-VIS absorption or photoluminescence emission spectroscopies, for instance, can hardly give any significant information regarding the size of the NWs or the type of crystal lattice. Therefore, this sentence should be better rewritten as:

--The as-grown tetra-diamond NWs have the same diameter distribution as pristine MWCNTs and have been characterized (rather than demonstrated) by high resolution transmission electron microscopy, fast Fourier transform, electron energy loss, ultraviolet-visible, and photoluminescence spectroscopies to have a diamond-like lattice with each carbon replaced by a carbon tetrahedron and a lattice constant of 7.80 Å.

On the other hand, while in the structural model of tetra-diamond shown in Fig. 2d each carbon atom has been replaced by a tetrahedron of carbons, in Fig. S5 showing the optimized structure by PAW method with PBE functional, has one additional tetrahedron of carbons in the center of the structure. What is the reason for this difference?

Results and Discussion:

Experimental errors or uncertainties for all the parameters experimentally determined or theoretically calculated should be included in the manuscript.

1st paragraph, pag 1:

"MWCNTs were prepared by a CVD method to have diameters around 10-20nm (-please, include the estimated length of the CVD synthesized MWCNTs, since the next phrase is related to length shortening-) and further shortened by a sonication method to improve their dispersion."

3rd paragraph, pag 2:

It is difficult to understand why the NWs are all surrounded by amorphous structures which were produced from methanol by laser irradiation, while the unreacted MWCNTs are not. What is the explanation for this fact? Is there any kind of covalent bond between the NWs and the amorphous structures? Please discuss this issue.

Regarding the electron energy loss spectroscopy (EELS) technique that was adopted to determine the elemental components of the as-produced NWs (Figure 1e). The small oxygen k-edge except from carbon k-edge, is it 25% calculated from the observed intensity ratios for O-K and C-K signals? A value of about 18% can be estimated from the graph in Fig 1e ! Please, specify this issue. What is the meaning/information obtained from this 25% concerning the composition of the sample?

On the other hand:

"Only electron transition from 1s level to σ^* band was observed from the carbon k-edge spectrum, where the electron transition from 1s level to π^* band is negligible (Figure 1e)."

Please, include an insert of the region involving the transition from 1s level to π^* band in Fig. 1e. From the results discussed at the end of this paragraph it can be assumed that the ratio of sp² carbon is lower for NWs (data not provided, could a rough estimation be made?) than for unreacted MWCNTs (51.6%) and for the amorphous structures (70.4%). What is the reason for the highest ratio of sp² carbon in the amorphous structures compared to MWCNTs.

1st paragraph, pag 3:

From the analysis of the FFT patterns of one single carbon NW at different tilting angles, the crystal lattice of the carbon NWs is described as tetra-diamond (please see the comment above just at the end of the comments concerning the Abstract). In addition, two distinct C-C bond lengths of 1.558 Å (intra-tetrahedron) and 1.470 Å (inter-tetrahedron) were determined. However, while the bond lengths of ca. 1.54 Å can be well ascribed to Csp³-Csp³ single bonds, those about 1.47 Å are typical of Csp²-Csp² single bonds. What is the explanation for this fact if the tetra-diamond is expected to be composed of Csp³ atoms only? Furthermore, what is the explanation for inter-tetrahedron distances shorter than those intra-tetrahedron? Is any reminiscent double bond character (from the MWCNTs) possible between the tetrahedrons of carbons? If the lattice optimized structure shown in Fig. S5 is considered, with every C atom having one inter-tetrahedron bond (i.e., some Csp² character) and three intra-tetrahedron bonds, should some homoconjugation be expected? What implications could have this fact and has it been taken into account in the theoretical calculations? Nothing regarding this issue is mentioned when the absorption and emission spectra are discussed.

Concerning the spectra shown in Fig 4, S4 and S6:

- The optical density (i.e., absorbance) has no units and, therefore, a.u. should be removed from the vertical axis in Fig. 4a. On the other hand, the vertical axis should show the right numbers (experimental absorbance values give important information about the amount of light absorbed by the solution, especially if irradiation light of 532 nm is used and the spectral line -red for the CNTs + methanol solution in Fig 4a, however, cut between 400 and 800 nm- is relatively high at 532 nm, how much???)

- The CNTs + methanol solution in Fig 4a has not negligible absorbances in the 200-1200 range, however, the absorption of the CNTs + methanol solution after laser irradiation is about twice in the VIS region. What is the reason for this increment in the optical density? Is it a property of the new NWs or of the amorphous structures? ("The solution after laser ablation of MWCNT suspension in methanol was directly used as UV-Vis and PL measurements."). Provided that no changes can be seen in the "Methanol by laser absorption spectrum" shown in Fig. 4a, compared with the absorption spectrum of methanol (negligible light absorption above 240 nm, its cut-off wavelength) can really the formation of amorphous structures be ascribed to production from methanol by laser ablation? If the amorphous structures have the highest ratio of sp² carbon (70.4% vs 51.6 % for MWCNTs) and MWCNTs have an absorption maximum near 250 nm, what is the reason why the amorphous structures apparently generated from methanol under laser irradiation do not show any absorption band above 240 nm? Could the amorphous structures be generated from MWCNTs as well, or even from the NWs (decomposition?) provided they intensely absorb light of 532 nm and the samples are irradiated for 1 hour with a repeating frequency of 1000 Hz and 75 W optical power?

- Regarding the emission spectra in Fig. 4b, S4 and S6, numerical values should be included in the vertical axes of Figs. 4b and S4. The sharp peaks that can be observed in the left side of the emission band (Figs 4b, S4 and S6) are due to the Raman band of the solvent and, therefore, should not be considered, especially if data from Fig. S4 are used in the calculation of emission quantum yields. How this value has been calculated, i.e., relative to an emission standard (which one?) or by means of an integrating sphere? And for what emission range (i.e., no Raman band considered)? Please, address these issues. On the other hand, a value of 5.41% (0.0541) for the emission quantum yield is excessively accurate and of little credibility. Again, the experimental uncertainty should be indicated together with the calculated value.

Fig. S4 caption:

"Red curve shows the blank solvent (methanol); black curve shows absorption ??? and emission from as-produced NWs." Please, remove the word absorption.

1st paragraph, pag 4:

- The diameter of the NWs, contrary to what is claimed by the authors, seems to vary with the radiation power (at 85 mW the highest % of population is in the 0-10 nm range, while at 75 or 95

mW the highest % of population is in the 10-20 nm range, unless, again, the experimental uncertainty is larger than expected).

- Regarding the statement that "The diameter distribution of the tetra-diamond NWs (Figure 5b) is well consistent with that of MWCNTs used for laser ablation, further confirming the success of the pseudo-totactic conversion of MWCNTs into tetra-diamond NWs."

If MWNTs with outer diameter in the 10-20 nm range are the starting material (assuming inner diameter of 5-15 nm typical of similar commercial products) and the distance between each wall is about 0.34 nm, i.e., 3.4 Å (graphite-like), an estimation of ca. 15-45 layers can be calculated. When the MWCNTs are transformed into NWs and the layers (originally separated by 3.41 Å and held together by weak dispersion forces) collapse into a 3D network of sp^3 atoms linked by bonds with ca. 1.5 Å bond lengths, a structural contraction in the NW could be expected when compared with the MWCNT precursor, maybe in agreement with diameter results at 85 mW. Could the authors comment on these considerations?

Fig. 6. What is the peak with retention time of ca. 2 min?

2nd paragraph, pag 4:

"The MWCNTs were transformed into tetra-diamond NWs under laser irradiation. The dangling bonds of sp^3 carbon were passivated by the hydrogen dissociated from methanol. The amorphous-like carbons further adhere to the surface of tetra-diamond NWs (Figure 1b) to act as a passivation layer to preserve the as-grown tetra-diamond NWs. The preservation of the shape of MWCNTs to tetra-diamond NWs might be due to the higher absorption of 532nm photons to MWCNTs than methanol solvent. The MWCNTs were multiphoton excited to high energy and transferred to tetra-diamond NWs and further trapped to room temperature."

The text above tells the story about NWs formation under pulsed-laser irradiation, however, several dark aspects are not clarified, for instance:

- The amorphous structures were generated from methanol or from decomposed NWs?
- Are the amorphous structures bound to the NWs by covalent bonds?
- Since methanol solvent does not absorb light at 532 nm (no absorption band for the solvent at this wavelength), what is the relationship between preservation of the shape of MWCNTs to tetra-diamond NWs and higher absorption of 532 nm photons by MWCNTs, especially if absorption at 532 nm by NWs is even higher for the NWs?
- Could photobleaching of the NWs be expected at long irradiation times, i.e., NWs not completely trapped to room temperature?

Minor comments:

- The term "pseudo-totactic" can be found several times in the manuscript, however, the term "pseudo-topotactic" should be used.
- The term "graphene nanoribbons" should substitute for "grapheme nanoribbons" in the 1st paragraph of the manuscript.

Reply to the Referees

The followings are the details of our replies to the comments and questions raised by the referees, where the referees' comments are shown in blue italic fonts for the sake of clarity. The change has been made to the manuscript and supporting information accordingly. The revised manuscript benefits significantly from the referees' thoughtful comments. In particular, we would like to report our point-to-point response to each comment from referees as follows.

Referee 1

Recommendation: A new carbon allotrope, i.e., T-carbon, with the structure formed by each carbon in cubic diamond replaced with a carbon tetrahedron, which was predicted by first-principles calculations several years ago, is now asserted to be obtained in the form of nanowires from carbon nanotubes in methanol under picosecond pulsed-laser ablation. The synthesis procedure is described, and the measurements and analyses of HRTEM, FFT, EELS, UV - Vis, and PL are presented, which appear to show, compatibly, the diamond-like structure “is consistent with a theoretically predicted T-carbon structure (lattice constant 7.52 Å)” . If the evidence is solid, this work would be a breakthrough in the field of carbon science, as it is the third identified three-dimensional periodic structure of carbon allotropes (the other two include cubic diamond and lonsdaleite). It will also deepen our further understanding on the diversity of carbon bonding. I would recommend publication of this manuscript provided that the following points are carefully clarified:

Point (1) of Referee 1:

I note that the authors use HRTEM and FFT to characterize the new structure. They only provide high-index FFT patterns. Can the low-index FFT pattern be given? Or can a direct electron diffraction image (other than FFT pattern) with low index on the new structure be provided? Such information will further convince the reader and strengthen the statement.

Answer:

‘Only lattice planes with either all even whose sum is a multiple of 4 or all odd indexes were observed in FFT patterns due to the extinction rules for T-carbon, which were demonstrated in supporting information.’ has been added to the end of the right column of page 3.

The low index FFT patterns corresponding to $\{100\}$ or $\{110\}$ are extinct.

The structure factors and extinction rules have been added in the page 2-3 of supporting information. Since the structure factor of T-carbon is shown in

$$F_{hkl} = f[1 + (-1)^{h+k} + (-1)^{h+l} + (-1)^{k+l}][1 + (-i)^{h+k+l}] \chi(a, h, k, l) \dots \dots \dots (S3)$$

- If h, k, l are of mixed parity (odd and even values combined) the first term is zero, so $|F_{hkl}|^2 = 0$
- If h, k, l are all even or all odd then the first term $[1 + (-1)^{h+k} + (-1)^{h+l} + (-1)^{k+l}] = 4$
- ✓ if h,k,l are all odd, $F_{hkl} = 4f(1 \pm i)$, then $|F_{hkl}|^2 = 32f^2$
- ✓ if h,k,l are all even whose sum is a multiple of 4 ($h + k + l = 4n$), then $F_{hkl} = 4f \times 2$, thus $|F_{hkl}|^2 = 64f^2$
- ✓ if h,k,l are all even whose sum is not a multiple of 4 ($h + k + l \neq 4n$), the second term is zero and thus $|F_{hkl}|^2 = 0$

Point (2) of Referee 1:

The shape of EELS for carbon (Fig. 1e) looks a bit strange. No low energy loss peak is observed, which may probably lead to a misjudgement on the ratio of sigma/pi orbitals.

Answer:

The EELS spectra were taken from the sample on carbon film. The background of the carbon film was subtracted from the EELS spectra before the calculation, resulting in the appearance of no low energy loss peaks. The calculation of sigma/pi ratio is believed to be more accurate with background correction.

‘Background correction was carried out for EELS spectra before the calculation of σ/π ratio to eliminate the effects of carbon film.’ has been added to the 4th paragraph of the right column in page 2.

‘after subtraction of carbon film background’ has also been added to the end of the caption of figure 1 and Figure S3.

The original figure and background subtraction process are shown in Figure I. The blue line is the measured signal, the red line is the background, and the green line is the signal after elimination of backgrounds.

Figure I. EELS spectrum of T-carbon nanowires surrounded by amorphous structures. The T-carbon is only composed of sp^3 carbon. Only electron transition from 1s level to σ^* band was observed from the carbon k -edge spectrum, where the electron transition from 1s level to π^* band is negligible (Figure 1e). That is why the lower energy loss corresponding to electron transition of 1s level to π^* band is disappeared compared to normal carbon K-edge.

Point (3) of Referee 1:

In the whole manuscript, “tetra-diamond” should be changed to “T-carbon”, as the latter is the widely accepted name for such a structure of carbon.

Answer:

‘tetra-diamond’ has been changed to ‘T-carbon’ throughout the manuscript and supporting information.

Point (4) of Referee 1:

Typos. In title: “Psuedo-topotactic” should be “Pseudo-topotactic”; pages 1 &4, several places, “tpotactic” should be “topotactic”; there are other typos throughout the text.

Answer:

‘psuedo’ and ‘tpotactic’ has been changed to ‘pseudo’ and ‘topotactic’, respectively throughout the manuscript and supporting information.

‘grapheme nanoribbons’ has been corrected to be ‘graphene nanoribbons’ in the 1st paragraph.

Referee 2

Recommendation: The authors report conversion of carbon nanotubes into tetra diamond nanowires by picosecond laser irradiation. Nanotubes are suspended in methanol and the conversion into tetra diamond is claimed by some pseudo-topotactic growth, because size distribution of nanotubes and tetra diamond nanrods remains the same. There are so many

technical loose ends which needs to be addressed and tied down firmly, before this work can be considered for publication.

Point (1) of Referee 2:

This conversion from nanotubes to tetra diamond involves a distinct change in structure (entropy), which implies it is a first-order phase transformation.

Answer:

We thank reviewer for pointing out this important issue. Yes. It is the first-order phase transformation.

‘The phonon density of states, specific heat, and entropy of graphite, diamond, CNT, and T-carbon were calculated with the quasi-harmonic approximation as shown in Figure S9 and Table S3.⁴³ The computed longitudinal optical (LO) of diamond (CNT (6,6)) is 35 (45) THz, which agrees nicely with the experimental measured value of 36 (44) THz by inelastic x-ray scattering.^{44,45, 46} The computed LO mode of T-carbon is 53 THz (1767 cm^{-1}), in a good agreement with reported value, 1760 cm^{-1} .²⁵ The LO mode of T-carbon has higher frequency than that of diamond is due to the strain stored in tetrahedrons. A distinct variation of entropy from the MWCNTs to T-carbon was involved in the pseudo-topotactic transformation of MWCNTs to T-carbon nanowires, indicating this transformation to be the first-order phase transformation.’ has been added to be the left column of page 5.

‘The phonon, entropy, and specific heat are computed with the quasi-harmonic approximation.⁴³’ has also been added to the end of the method section in page 6.

The methods for quasi-harmonic approximation calculations have been added to page 3-4 of supporting information.

List of specific heat (Cp) and entropy (S) for the graphite, diamond, CNT(6,6) and T-carbon has been added to supporting information as table S3.

Phonon density of states (Phonon-DOS) of diamond, graphite, T-carbon and CNT(6,6) has been added to supporting information as Figure S9.

Point (2) of Referee 2:

What is the template for pseudo-topotactic growth and resulting first-order phase transformation?

Answer:

‘Template growth of 1D carbon nanostructures (CNTs,^{8, 9} carbon linear chain,^{10, 11} diamond nanowires (NWs),^{12, 13} graphene nanoribbons^{5, 6}) in the inner chambers of CNTs has been

well studied. However, the pseudo-topotactic conversion of CNTs into carbon NWs has not yet been reported.’ has been explained in the first paragraph.

MWCNTs serve as auto templates for T-carbon transformation.

Point (3) of Referee 2:

Why NW samples were destroyed under Selected-area-electron-diffraction (SAED), which should give more definitive results than FFT patterns on the structural characterization.

Answer:

Yes, SAED can provide more definitive results than FFT patterns on the structure characterization. Some tetra-diamond nanowires are destroyed under electron flux of 6.8×10^4 [e]/(nm²·s) for more than 10s. The electron flux for SAED measurement is several magnitudes larger than the above one. The adjustment time is usually more than 1min. The tetra-diamond nanowires are destroyed by such a large amount of electron flux. Carbon nanowires such as diamond nanowires were also found to be damaged by electron flux of 1.7×10^5 [e]/(nm²·s) (*Angew. Chem. Int. Ed.* 2013, 52 (13), 3717)

‘FFT patterns were used instead of selected area electron diffraction (SAED) for structure analysis since the NW was destroyed during SAED measurements’

has been replaced by

‘FFT patterns were used instead of selected area electron diffraction (SAED) for structure analysis since the NWs were unstable under SAED measurement conditions where an electron flux of several orders of magnitude larger than 10^5 [e]/(nm² · s) was used.’ in the left column of page 2.

Point (4) of Referee 2:

In reviewer’s opinion, nanotubes are being melted under a super undercooled state and converted into diamond by the mechanism proposed in recent exciting breakthroughs (J. Appl. Phys 118, 215303(2015)).

Answer:

The solid sample of amorphous carbon film was used as precursors in J. Appl. Phys 118, 215303(2015)). The formation of nanodiamond is a result of homogeneous nucleation of diamond phase from the highly undercooled pure carbon in the absence of a template. The laser pulse width used for irradiation is nanosecond.

The carbon nanotubes in our work serve not only as precursors but also auto-templating structural transformation. The carbon nanotube precursors suspended in methanol solvent

instead of solid films were used for laser irradiations. The pulse width used in our work for irradiation is picosecond laser. The picosecond laser interaction with materials is different from that of nanosecond laser. Nanosecond laser interaction often generates more heat. However, the interaction of picosecond laser with materials is usually multi-photon excitation which caused the structure transformation.

‘The overall shape of CNT auto templates was preserved to yield NW products instead of nanoparticles by irradiation with picosecond pulsed laser instead of nanosecond one. Much stronger heating effect of nanosecond laser than that of picosecond one results in the products with different morphology. The MWCNTs were multi-photon excited to high energy and transferred to T-carbon NWs and further trapped to room temperature. The mechanism of structure transformation of MWCNTs to T-carbon is believed to be different from that proposed for the transformation of amorphous carbon film to diamond induced by nanosecond laser.’⁴⁷ has been added to the 3rd paragraph of page 5 (left column).

The reference J. Appl. Phys 118, 215303(2015) has been cited.

Referee 3

Recommendation: The authors claim to have synthesized tetra-diamond carbon nano-wires (NWs) by irradiating multi-walled carbon nanotubes (MWCNTs) with a pico-second laser. The claim represents a novelty regarding the use of pico-second laser and the experimental identification of the tetra-diamond carbon phase. But there are comments from experimental view that need to be addressed:

Point (1) of Referee 3:

The authors chiefly support the synthesis of the tetra-diamond NW by matching the measured lattice constant of the laser-synthesized material to the theoretical predictions of the reference 25. For that, the experimental measurement is obtained from Fast Fourier (FFT) generated from high resolution transmission electron microscopy (HREM) images. Nevertheless, neither comparisons between the FFT of the tetra-diamond NW with NWs existing in different phase nor with the starting MWCNT are provided. This does not permit clearly separate the tetra-diamond phase from other products that can also be created by the laser treatment.

Answer:

‘Lots of NWs (Figure S2) except from some unreacted MWCNTs (Figure S3a) and amorphous structures (Figure S3b) were observed by HRTEM from the solution after laser irradiation.’

has been replaced by

Many unreacted MWCNTs and amorphous phases in addition to NWs were observed by HRTEM from the suspension after laser irradiation (Figure S2). The NWs were observed to have same crystal planes and interplanar crystal spacings, which are distinguishable from unreacted MWCNTs (Figure 1a&S3a) and amorphous phases (Figure S3b),’ in the left column of page 2. Figure S2 has also been replaced.

The structures and FFT pattern of starting MWCNTs have been added to the insets of figure 1a.

‘(insets show the structure and FFT pattern of pristine MWCNTs)’ has also been added to the corresponding caption of Figure 1.

Point (2) of Referee 3:

Electron energy loss spectroscopy (EELS) is used to quantify the sp^3 bonds since the reference 25 predicts massive presence of such bonds in the tetra-diamond phase. However, this EELS discussion in the manuscript lies mostly on amorphous and MWCNT forms and it is unclear the EELS information from the laser-created NW.

Answer:

The NWs are always covered by amorphous carbon and TEM observation was carried out on a holey carbon grid which led to a large background sp^2 signals. However, the signal of sp^3 is unambiguous. To make it more clear, the calculation of sp^3 hybridized carbon has been reorganized in the 5th paragraph of page 2.

‘Only electron transition from 1s level to σ^* band was observed from the carbon k -edge spectrum, where the electron transition from 1s level to π^* band is negligible (Figure 1e). The $[\pi^*/(\pi^*+\sigma^*)]$ ratio of the NWs was estimated to be 0.04 ± 0.01 . The sp^2 ratio was calculated to be $16\pm 4\%$ due to the surrounding amorphous structures. The carbon NWs are supposed to be composed of only sp^3 carbon, which is similar to diamond.³⁵ The $[\pi^*/(\pi^*+\sigma^*)]$ ratio of the unreacted MWCNTs (Figure S3a) was estimated to be 0.13 ± 0.01 according to the carbon K -edge (Figure S3c). The ratio of sp^2 carbon is calculated to be $51.6\pm 0.03\%$.^{13, 36} The high sp^3 ratio of the NWs is consistent with the T-carbon structure.’ has been explained in the 5th paragraph of page 2.

‘Electron energy loss spectroscopy (EELS) was adopted to determine the elemental component of the as-produced NWs (Figure 1e).’

has been replaced by

‘Electron energy loss spectroscopy (EELS) was adopted to determine the valence, chemical bonding, and atomic composition of the as-produced NWs (Figure 1e).³⁴’ in the beginning of 5th paragraph of page 2.

The relevant reference has been cited.

‘It is very clear that EELS analysis has shown that sp² hybridization in MWCNTs has been transformed to sp³ hybridization. And furthermore, the amorphous coating on the surface of NWs contained oxygen, which is from methanol conversion.’ has been added to the end of the 5th paragraph of page 2.

Point (3) of Referee 3:

In the reference 25, it is predicted 3.0 eV band gap for the new carbon phase. In the manuscript, it is shown, from UV-Vis absorption, bands corresponding to 5.51, 4.96 and 4.49 eV. Hence, the measurements are inconsistent with the predictions.

Answer:

DFT band gap varies greatly depend on functional used. We have used a more accurate G₀W₀ method (provides more accurate description of electronic structure) to calculate the band structure again. The bandgap obtained is 5.10 eV, in consistent with UV-Vis absorption. The G₀W₀ corrected band structure together with the projected density of states were used to update Figure 3 in the revised manuscript.

‘The tetra-diamond structure exhibits a direct band gap of 3.4 eV (Figure 3) with optimized lattice constant of 7.49 Å (Figure S5). The exciton binding energy was computed to be 180 meV which is over two times larger than that of diamond (i.e. 80 meV⁴²), while is much smaller than those in carbon nanotubes,⁴³ graphene,⁴⁴⁻⁴⁶ and graphene nanoribbons.⁴⁷⁻⁴⁹ The electronic band structures and density of states (DOS) of tetra-diamond are well consistent with the ultraviolet-visible (UV-Vis) absorption and photoluminescence (PL) of the products (Figure 4).’

has been updated to

‘The subsequent G₀W₀ corrected band structure with direct bandgap of 5.10 eV at G point and projected density of states (PDOS) are presented in Figure 3. The valence band maximal and conduction band minimal exhibit p-orbital character. The electronic band structures and

PDOS of T-carbon provide basis to corroborate the ultraviolet-visible (UV-Vis) absorption and photoluminescence (PL) of the products (Figure 4).’ in the 1st paragraph of page 4 (left column);

and

‘With the computed gap (5.10 eV) and the first absorption peak 4.54 eV (273 nm) as shown in the inset of Figure 4a, the exciton binding energy is 560 meV, which is eight times exciton binding affinity of diamond (70 meV).⁴² The strong exciton binding energy enables the observation of exciton feature in optical measurements at room temperature. The inset in Figure 4(a) shows the computed absorption spectrum with pronounced peaks at 5.48 eV (226 nm), 5.17 eV (240 nm), and 4.54 eV (273 nm) in a good agreement with experimental measured absorption spectrum.’ in the 1st paragraph (left column) of page 4.

Point (4) of Referee 3:

A minor issue. A Nd:YAG laser has wavelength of 1064 nm that can be doubled in frequency to produce 532 wavelength laser beam. The authors should consider this at the third paragraph in the end of the first page when describing the methods.

Answer:

‘A Q-switched Nd:YAG laser with a wavelength of 532 nm’ has been replaced by ‘A Q-switched laser with a wavelength of 532 nm (doubled from a Nd:YAG laser with a wavelength of 1064 nm),’ in the left column of page 2 and method section in page 6.

Point (5) of Referee 3:

The manuscript has some typos and needs to be reviewed with respect to proper use of the writing:

- ①. *In the abstract, the second sentence has the typo ‘pseudo-tpotactic’.*
- ②. *The term ‘pulse width’ for the temporal dynamics of the laser pulse is not commonly used. Instead, usually one writes time duration. Consider revising this at the third paragraph in the end of the first page.*
- ③. *To a more proper formal language, it should be considered alternative terms for ‘lots of’ and ‘it is hard’.* Both terms appear in the fourth paragraph.
- ④. *In the first sentence of the sixth paragraph, ‘has’ must be changed to have since its subject is NWs in the plural.*
- ⑤. *In the end of the eighth paragraph, change ‘The PL intensity increase’ to The PL intensity increases since subject is third person.*

⑥. In the end of the tenth paragraph, 'NWs might due to' should be changed to 'NWs might be due to'.

⑦. From a broader view, MWCNTs have been submitted to laser irradiation as in del Pino, J. Appl. Phys, 2014. Although a nano-second laser was used in this work, its conclusions indicate the formation of different phases of carbon. Thus, such a work and related ones should be mentioned and cited.

Answer:

① 'pseudo-tptotactic' in the second sentence of abstract has been corrected to be 'pseudo-topotactic'.

② 'pulse width' has been corrected to be 'pulse duration' in the left column of second page.

③ 'lots of' has been corrected to be 'many' and 'It is hard to tilt the NWs to other desired zone axes' has been replaced by 'Tilting the NWs to other desired zone axes is a big challenge' in the fourth paragraph.

④ 'has' has been corrected to be 'have' in the first sentence of the sixth paragraph.

⑤ 'The PL intensities increase' has been corrected to be 'The PL intensity increases' in the end of the eighth paragraph.

⑥ 'NWs might due to' has been corrected to be 'NWs might be due to' in the end of the tenth paragraph.

⑦ 'Different structural forms of carbon were also produced by laser irradiation of CNT films under nitrogen atmosphere.³⁰' has been added to the end of 1st paragraph in page 2. The paper of del Pino, J. Appl. Phys, 2014. has been cited.

Point (6) of Referee 3:

In general, the comments 1-4 pose important questionings on the experimental evidences that state the synthesis of the tetra-diamond NW using the laser method. And, in this stage, these issues represent strong inconsistencies. If the authors can convincingly provide more experimental evidences from the data in the manuscript, maybe, Raman measurements could be performed, and/or clarify those comments, a further review round should be considered.

Answer:

The comments 1-4 have been clarified and extra data have been added.

Raman measurements of sample have been performed and added to supporting information as Figure S8.

‘Raman spectrum of the products after laser irradiation was obtained by using an excitation laser of 514 nm (Figure S8, black), which is distinguishable from that of diamond structures (Figure S8, red).’ has been added to the end of 1st paragraph of page 4 (right column).

‘Raman spectroscopy was taken in a back-scattering geometry using a single monochromator with a microscope (Reinishaw inVia) equipped with CCD array detector (1024 × 256 pixels, cooled to 570 °C) and an edge filter. The samples were excited by 514.5 nm Argon ion laser.’ has been added to the method section in page 6.

Referee 4

Recommendation: The authors report on the pseudo-topotactic conversion of carbon nanotubes to tetra-diamond nanowires in methanol under irradiation of picosecond laser and claim the formation of a new carbon allotrope. However, based on the following considerations, this reviewer does not feel that the aforementioned goal has been achieved in an unequivocal manner and, since reasonable doubts are in the manuscript, I do not consider that this work meets the criteria for publication in Nat. Commun.

Answer:

Before respond each comment and question, we want to express our sincere appreciation to the referee for his or her professionalism, time and most importantly insightful comments and questions. We have never received a review comparable to this in seriousness, in thoroughness and in depth. As the referee knows, our process involves three phases of solid, liquid and gas and takes place under a highly kinetically controlled condition. There are many questions to be asked. The referee’s comments and questions touched a quit bit of them. To answer them unequivocally requires much more study well beyond the scope of this manuscript. We try our best to show that our evidence is sufficient to prove that T-carbon previously predicated has been successfully prepared and observed for the first time.

Point (1) of Referee 4:

Abstract:

This reviewer wonders whether the term “laser ablation ” is suitable to describe the preparation of the carbon NWs, since no material is removed from a solid or liquid surface but transformed with a laser beam. Maybe “picosecond pulsed-laser irradiation ” is more suitable and accurate..

Answer:

‘laser ablation’ has been replaced by ‘laser irradiation’ throughout the manuscript and supporting information.

Point (2) of Referee 4:

① Concerning the text:

The as - grown tetra - diamond NWs have the same diameter distribution as pristine MWCNTs and been demonstrated by high resolution transmission electron microscopy, fast Fourier transform, electron energy loss, ultraviolet - visible, and photoluminescence spectroscopies to have a diamond - like lattice with each carbon replaced by a carbon tetrahedron and a lattice constant of 7.80 Å. ”

Techniques such as UV-VIS absorption or photoluminescence emission spectroscopies, for instance, can hardly give any significant information regarding the size of the NWs or the type of crystal lattice. Therefore, this sentence should be better rewritten as: --The as - grown tetra - diamond NWs have the same diameter distribution as pristine MWCNTs and have been characterized (rather than demonstrated) by high resolution transmission electron microscopy, fast Fourier transform, electron energy loss, ultraviolet - visible, and photoluminescence spectroscopies to have a diamond - like lattice with each carbon replaced by a carbon tetrahedron and a lattice constant of 7.80 Å.

② *On the other hand, while in the structural model of tetra - diamond shown in Fig. 2d each carbon atom has been replaced by a tetrahedron of carbons, in Fig. S5 showing the optimized structure by PAW method with PBE functional, has one additional tetrahedron of carbons in the center of the structure. What is the reason for this difference?*

Answer:

① ‘The as-grown tetra-diamond NWs have the same diameter distribution as pristine MWCNTs and been demonstrated by high resolution transmission electron microscopy, fast Fourier transform, electron energy loss, ultraviolet–visible, and photoluminescence spectroscopies to have a diamond-like lattice with each carbon replaced by a carbon tetrahedron and a lattice constant of 7.80 Å.’

has been replaced by

‘The as-grown T-carbon NWs have the same diameter distribution as pristine MWCNTs and have been characterized by high resolution transmission electron microscopy, fast Fourier transform, electron energy loss, ultraviolet–visible, and photoluminescence spectroscopies to

have a diamond-like lattice with each carbon replaced by a carbon tetrahedron and a lattice constant of 7.80 Å.’ in the abstract.

② The structure shown in original Figure S5 is the same with Figure 2d, only the coordination was paralleled moved. The Figure S5 has been deleted to avoid misunderstanding.

Point (3) of Referee 4:

Results and Discussion:

Experimental errors or uncertainties for all the parameters experimentally determined or theoretically calculated should be included in the manuscript.

Answer:

All the errors or uncertainties for the experimental parameters have been added. Since the theoretical methods are of deterministic type, that is, when the proper conditions are chosen, the results are fixed. Therefore, our focus on theoretical part is on methodology. For instance, the G_0W_0 method employed in this revised manuscript is a better method to compute band structure and band gap.

Point (4) of Referee 4:

1st paragraph, pag 1:

“MWCNTs were prepared by a CVD method to have diameters around 10 - 20nm (-please, include the estimated length of the CVD synthesized MWCNTs, since the next phrase is related to length shortening-) and further shortened by a sonication method to improve their dispersion.”

Answer:

‘MWCNTs were prepared by a CVD method to have diameters around 10-20nm^{31, 32} and further shortened’

has been replaced by

‘MWCNTs were prepared by a CVD method to have diameters around 10-20 nm^{31, 32} and lengths of dozens of micrometers and subsequently shortened’ in the 2nd paragraph of page 2.

Point (5) of Referee 4:

3rd paragraph, pag 2:

It is difficult to understand why the NWs are all surrounded by amorphous structures which were produced from methanol by laser irradiation, while the unreacted MWCNTs are not. What is the explanation for this fact? Is there any kind of covalent bond between the NWs and the amorphous structures? Please discuss this issue.

Answer:

This is an excellent question. The process is highly localized (point of laser interaction) and kinetically controlled. The fact that we have observed amorphous structure formation by irradiation of methanol and oxygen signal from NWs made us believe the amorphous structure on the surface came from methanol conversion, which could also come from the conversion of the surface carbon atoms of MWCNT, but the contribution is small because statistics information of T-carbon NW size suggested addition of the materials. The formation of amorphous structure on the surface of the NWs, but not on MWCNTs is related to the nature of the reaction – only those CNTs with sufficient light absorption were turned really “hot”, with enough to lead structural conversion to take place, the “hot” surface is highly reactive for amorphous structure bonding. It is likely that the “hot” surface “catalyzed” methanol conversion to amorphous structures and its formation at the very moment of the structural conversion of MWCNTs likely helped to stabilize meta-stable T-carbon structure. HRTEM images (Fig. 2) suggested that the formation of covalent bond between the NWs and the amorphous structures.

‘Covalent bonds might be produced between amorphous structures and NWs to passivate the T-carbon NWs during laser irradiation.’

has been added after

‘the NWs are all surrounded by amorphous structures which might be produced from methanol by laser irradiation.’ in the right column of page 2 to explain why the NWs are all surrounded by amorphous structures.

‘Small oxygen *k*-edge (25%) except from carbon *k*-edge was also obtained, which is attributed to the surrounding amorphous structures.’

has been replaced by

‘The small oxygen *k*-edge can be attributed to the surrounding amorphous structures produced from laser interaction with methanol because methanol is only oxygen source in the reaction system. To further confirm that the oxygen signal is related to the methanol conversion, methanol solvent have been irradiated under the same conditions. Large amount of the same amorphous structures were produced.’ In the right column of page 2.

Point (6) of Referee 4:

Regarding the electron energy loss spectroscopy (EELS) technique that was adopted to determine the elemental components of the as - produced NWs (Figure 1e). The small oxygen

k - edge except from carbon k - edge, is it 25% calculated from the observed intensity ratios for O-K and C-K signals? A value of about 18% can be estimated from the graph in Fig 1e ! Please, specify this issue. What is the meaning/information obtained from this 25% concerning the composition of the sample?

Answer:

The peak intensity ratio of oxygen to carbon is 0.18 ($\sim 1.5 \div 8.5$). However, the oxygen content is calculated by the measurement software according to the peak areas corrected by the different element factor because the k-edge absorption intensity of different element varies significantly from each other. The element content can not be simply judged directly from the ratio of their K-edge areas or peak intensities.

The oxygen might come from some ethers produced laser irradiation of methanol.

‘which is attributed to the surrounding amorphous structures’ has been replaced by ‘which is attributed to the surrounding amorphous structures produced from laser irradiation of methanol’ in the right column of page 2.

Point (7) of Referee 4:

On the other hand:

“Only electron transition from 1s level to σ^* band was observed from the carbon k - edge spectrum, where the electron transition from 1s level to π^* band is negligible (Figure 1e).”

Please, include an insert of the region involving the transition from 1s level to π^* band in Fig. 1e.

From the results discussed at the end of this paragraph it can be assumed that the ratio of sp² carbon is lower for NWs (data not provided, could a rough estimation be made?) than for unreacted MWCNTs (51.6%) and for the amorphous structures (70.4%). What is the reason for the highest ratio of sp² carbon in the amorphous structures compared to MWCNTs.

Answer:

The region involving the transition from 1s level to π^* and σ^* band have been inserted into Fig. 1e according to reference J. Micros. **2008**, 231, 144-155.

‘The $[\pi^*/(\pi^*+\sigma^*)]$ ratio of the NWs was estimated to be 0.04 ± 0.01 . The sp² ratio was calculated to be $16\pm 4\%$ due to the surrounding amorphous structures.’ has been added to the right column of page 2. The ratio of sp² carbon is estimated.

The carbon component of the amorphous structures is relative low, as shown in figure S3d. The measurement error of $[\pi^*/(\pi^*+\sigma^*)]$ is quite high. So the calculation error of sp^2 ratio is very high. The EELS of amorphous structures was used to compare the oxygen components. 'While the $[\pi^*/(\pi^*+\sigma^*)]$ ratio of the amorphous structures (Figure S3b,d) was estimated to be 0.18 ± 0.04 and the ratio of sp^2 carbon is calculated to be $70\pm 15\%$.' has been deleted from the 4th paragraph of page 2.

Point (8) of Referee 4:

1st paragraph, pag 3:

From the analysis of the FFT patterns of one single carbon NW at different tilting angles, the crystal lattice of the carbon NWs is described as tetra - diamond (please see the comment above just at the end of the comments concerning the Abstract). In addition, two distinct C-C bond lengths of 1.558 Å (intra-tetrahedron) and 1.470 Å (inter-tetrahedron) were determined. However, while the bond lengths of ca. 1.54 Å can be well ascribed to Csp3-Csp3 single bonds, those about 1.47 Å are typical of Csp2-Csp2 single bonds.

① *What is the explanation for this fact if the tetra-diamond is expected to be composed of Csp3 atoms only?*

② *Furthermore, what is the explanation for inter-tetrahedron distances shorter than those intra-tetrahedron?*

③ *Is any reminiscent double bond character (from the MWCNTs) possible between the tetrahedrons of carbons?*

④ *If the lattice optimized structure shown in Fig. S5 is considered, with every C atom having one inter-tetrahedron bond (i.e., some Csp2 character) and three intra-tetrahedron bonds, should some homoconjugation be expected?*

⑤ *What implications could have this fact and has it been taken into account in the theoretical calculations? Nothing regarding this issue is mentioned when the absorption and emission spectra are discussed.*

Answer:

① In T-carbon structure, each carbon has 4 covalent bonds with adjacent carbon atoms. Thus, there is no sp^2 carbon in this structure. The intra-tetrahedron bonds are largely deformed due to the tetrahedron. Consequently, the charge population analysis reveals that the electron density around the inter-tetrahedron bonds is higher than that on the intra-tetrahedron bonds as shown in the charge density plot in following figure:

T-carbon

Thus, the inter-tetrahedron bonds are much stronger than intra-tetrahedron ones, which plays an important role in stabilizing the whole structure by effectively balancing the strain from tetrahedron cages.

② The bond length of the inter-tetrahedron is shorter than that of the intra-tetrahedron due to the electron transfer from intra-tetrahedron bonds to inter-tetrahedron ones, as shown in the above figure.

③ There is no sp^2 carbon inside this structure. Only sp^3 carbons exist in this structure.

④ Since only sp^3 carbons exist in this structure, no homo-conjugation is anticipated.

⑤ To provide more direct evidence of bonding character, absorption spectra of T-carbon, graphene nanoribbon (sp^2 -GNR with width of W14), single-walled CNT (8,0) (sp^2 -CNT) and diamond (sp^3 -diamond) are computed and included as Figure S7. The bonding character has been clearly exhibited by the first absorption peak. The first transition peaks for GNR and CNTs lie at 1.5 eV, indicating clear sp^2 character. While the first transition peak of diamond lies at 7.0 eV, indicating definite sp^3 character. The first absorption peak of T-carbon (4.6 eV) and its shape resemble more of diamond's absorption spectrum than sp^2 ones. Therefore, carbon in T-carbon is sp^3 type rather than sp^2 one.

‘Although, similar to diamond structure, each carbon in T-carbon is covalently bonded with four adjacent carbon atoms via sp^3 hybridization, a localization in a group of four carbon atoms takes place which leads to substantial deviation of C-C bonding from the ideal sp^3 bonding in diamond structure. This deviation results in an appreciable electron transfer from intra-tetrahedron bonds to inter-tetrahedron ones. Therefore, the electron density of inter-tetrahedron bonds is much higher than that of intra-tetrahedron bonds, which leads to shorter bonds of inter-tetrahedrons than that of intra-tetrahedrons. The inter-tetrahedron bonds are much stronger than intra-tetrahedron ones, a distinctive structural feature in comparison to diamond structure.’ has been added to the end of paragraph 1 in page 3.

and

‘Moreover, the shape of the absorption spectrum of T-carbon shares more features of the absorption spectrum of diamond, with respect to that of spectra of graphene nanoribbon and single-walled CNT as shown in Figure S7. The first transition peak of T-carbon (4.6 eV) is also closer to that of diamond (7.0 eV) than that of graphene nanoribbon and single-walled CNT (1.5 eV), further confirmed that T-carbon is composed of sp^3 rather than sp^2 carbon.’

has been added in the 1st paragraph (right column) of page 4.

Point (9) of Referee 4:

Concerning the spectra shown in Fig 4, S4 and S6:

①- *The optical density (i.e., absorbance) has no units and, therefore, a.u. should be removed from the vertical axis in Fig. 4a. On the other hand, the vertical axis should show the right numbers (experimental absorbance values give important information about the amount of light absorbed by the solution, especially if irradiation light of 532 nm is used and the spectral line -red for the CNTs + methanol solution in Fig 4a, however, cut between 400 and 800 nm- is relatively high at 532 nm, how much???)*.

Answer:

① The UV-vis absorption shown in Fig. 4a were vertical shifted for comparison. ‘(absorption lines were vertical shifted)’ has been added to the caption of figure 4a. The baseline were added to Fig. 4a. The unit ‘a.u’ has been removed from the vertical axis in Fig. 4a. The values of the vertical axis have been added. The break was used between 400-800nm since no distinguishable absorption was obtained for the NWs. The full scale spectra without vertical shifts are shown in Figure S4.

‘The full scale absorption spectra are shown in Figure S4.’ has been added to the left column of page 4.

‘The preservation of the shape of MWCNTs to tetra-diamond NWs might due to the higher absorption of 532nm photons to MWCNTs than methanol solvent’

has been replaced by

‘The preservation of the shape of MWCNTs to T-carbon NWs might be due to the higher absorption of 532 nm photons by MWCNTs than that of NWs, where the absorption for MWCNT suspension is more than twice as that for NW or methanol solvent at 532 nm (Figure S4).’ at the right column of page 5.

Point (10) of Referee 4:

- The CNTs + methanol solution in Fig 4a has not negligible absorbances in the 200-1200 nm range, however, the absorption of the CNTs + methanol solution after laser irradiation is about twice in the VIS region. What is the reason for this increment in the optical density? Is it a property of the new NWs or of the amorphous structures? (“The solution after laser ablation of MWCNT suspension in methanol was directly used as UV - Vis and PL measurements. ”). Provided that no changes can be seen in the “Methanol by laser absorption spectrum ” shown in Fig. 4a, compared with the absorption spectrum of methanol (negligible light absorption above 240 nm, its cut-off wavelength) can really the formation of amorphous structures be ascribed to production from methanol by laser ablation? If the amorphous structures have the highest ratio of sp² carbon (70.4% vs 51.6 % for MWCNTs) and MWCNTs have an absorption maximum near 250 nm, what is the reason why the amorphous structures apparently generated from methanol under laser irradiation do not show any absorption band above 240 nm? Could the amorphous structures be generated from MWCNTs as well, or even from the NWs (decomposition?) provided they intensely absorb light of 532 nm and the samples are irradiated for 1 hour with a repeating frequency of 1000 Hz and 75 W optical power?

Answer:

The absorption of the CNTs + methanol solution after laser irradiation in the Vis region is less than half of the sample before irradiation (Figure S4), which is due to the transformation of MWCNTs into NWs.

The optical density of the sample after laser irradiation is not increased, as shown in Figure S4. The misleading of Figure 4a has been clarified and baselines have been added to Figure 4a.

The same amorphous structures were largely produced by laser irradiation of methanol. So the amorphous structures are assumed to be from laser irradiation of methanol.

‘which might be produced from laser irradiation of methanol’ has been added after ‘amorphous structures’ in the right column of page 2.

The sample was multi-photon excited and further converted to different phases, which was explained in the left column of page 5. The sample absorption at 532nm might not be directly related to the structure transformations.

‘Small oxygen *k*-edge (25%) except from carbon *k*-edge was also obtained, which is attributed to the surrounding amorphous structures.’

has been replaced by

‘The small oxygen *k*-edge can be attributed to the surrounding amorphous structures produced from laser interaction with methanol because methanol is only oxygen source in the reaction system. To further confirm that the oxygen signal is related to the methanol conversion, methanol solvent have been irradiated under the same conditions. Large amount of the same amorphous structures were produced.’ in the right column of page 2.

Yes, the amorphous structures could be generated from MWCNTs as well, but the result of diameter statistics analysis suggested materials has been added to NWs, therefore the amorphous coating is mainly from methanol conversion. Thought carbon hybridization in amorphous structure is believed to be mainly sp² – the same hybridization of carbon atoms in the MWCNTs, lack of absorption above 240 nm may be due to its disorder (low level of conjugation) and oxygen inclusion.

The ratio of sp² carbon has been clarified in **Point (7) of Referee 4**.

Point (11) of Referee 4:

- Regarding the emission spectra in Fig. 4b, S4 and S6, numerical values should be included in the vertical axes of Figs. 4b and S4. The sharp peaks that can be observed in the left side of the emission band (Figs 4b, S4 and S6) are due to the Raman band of the solvent and, therefore, should not be considered, especially if data from Fig. S4 are used in the calculation of emission quantum yields. How this value has been calculated, i.e., relative to an emission standard (which one?) or by means of an integrating sphere? And for what emission range (i.e., no Raman band considered)? Please, address these issues. On the other hand, a value of 5.41% (0.0541) for the emission quantum yield is excessively accurate and of little credibility. Again, the experimental uncertainty should be indicated together with the calculated value.

Answer:

The numerical values have been included in the vertical axes of Figure. 4b and S5 (original S4), S6. Figure 4b and Figure S5 have been replaced. The Raman band of methanol has been pointed out in Figure 4b. The absolute emission scales of samples have been shown in the figures. The calculation methods have been demonstrated in Fig S5 and its captions. The Raman band of solvent has been removed from the calculation of quantum yield, as shown in Figure S5. The calculation error has been added.

Point (12) of Referee 4:

Fig. S4 caption:

“Red curve shows the blank solvent (methanol); black curve shows absorption ??? and emission from as - produced NWs.” Please, remove the word absorption.

Answer:

The caption of original Fig. S4 (now Figure S5) has been replaced by ‘Quantum yield measurement of the T-carbon NWs. The quantum yield is calculated as the ratio of the number of photons emitted (N^{em}) by the number of photons absorbed (N^{abs}). Number of emitted photons is given by the area under the spectrally corrected emission (A^{em}). Number of absorbed photons is given by the difference of areas under the Rayleigh scattering peaks of a reference sample and a sample under study ($N^{\text{abs}}=A^{\text{scat}}_{\text{ref}}-A^{\text{scat}}_{\text{sample}}$). $Q= N^{\text{em}}/ N^{\text{abs}}= A^{\text{em}}/(A^{\text{scat}}_{\text{ref}}-A^{\text{scat}}_{\text{sample}})=5.4\pm 0.2\%$ ’.

Point (13) of Referee 4:

1st paragraph, pag 4:

- The diameter of the NWs, contrary to what is claimed by the authors, seems to vary with the radiation power (at 85 mW the highest % of population is in the 0-10 nm range, while at 75 or 95 mW the highest % of population is in the 10-20 nm range, unless, again, the experimental uncertainty is larger than expected).

- Regarding the statement that “The diameter distribution of the tetra - diamond NWs (Figure 5b) is well consistent with that of MWCNTs used for laser ablation, further confirming the success of the psuedo - ptotactic conversion of MWCNTs into tetra - diamond NWs.”

If MWNTs with outer diameter in the 10-20 nm range are the starting material (assuming inner diameter of 5-15 nm typical of similar commercial products) and the distance between each wall is about 0.34 nm, i.e., 3.4 Å (graphite-like), an estimation of ca. 15-45 layers can be calculated. When the MWCNTs are transformed into NWs and the layers (originally separated by 3.41 Å and held together by weak dispersion forces) collapse into a 3D network of Csp3 atoms linked by bonds with ca. 1.5 Å bond lengths, a structural contraction in the NW could be expected when compared with the MWCNT precursor, maybe in agreement with diameter results at 85 mW. Could the authors comment on these considerations?

Answer:

‘The diameters of the tetra-diamond NWs are mainly distributed between 10 to 20 nm, not affected by laser power between 75 to 95 mW (Figure 5b). The diameter distribution of the tetra-diamond NWs (Figure 5b) is well consistent with that of MWCNTs used for laser

ablation, further confirming the success of the pseudo-topotactic conversion of MWCNTs into tetra-diamond NWs.’

has been replaced by

‘The diameters of the T-carbon NWs are mainly distributed between 10 to 20 nm, with average diameter of 11.8 ± 2.8 nm (Figure 5b). The diameter distribution of the T-carbon NWs (Figure 5b) is well consistent with that of shortened MWCNTs (11.7 ± 2.2 nm) used for laser irradiation, further confirming the successful pseudo-topotactic conversion of MWCNTs into T-carbon NWs.’ at the beginning of page 5.

An excellent point about diameter contraction. The length analysis showed the diffusion of carbon atoms during phase transition was largely limited to the restriction in perpendicular to the axis of the MWCNTs (no obvious length change), a diameter contraction is inevitable. However statistics analysis of diameters of NWs showed that the diameter distribution of the NWs is similar to that of the starting MWCNTs. This fact plus amorphous structure formation by irradiation of pure methanol, as above discussed, made us believe that materials – amorphous structure from methanol conversion has been added to the surface of the NWs, which obscured the observation of the diameter contraction.

Point (14) of Referee 4:

Fig. 6. What is the peak with retention time of ca. 2 min?

Answer:

The peak with retention time about 2 min are due to the disturbances of rotary switch in the equipments.

The peak has been pointed out in the Figure 6 and clarified in the caption as ‘(* from the disturbance of equipment)’.

Point (15) of Referee 4:

2nd paragraph, pag 4:

“The MWCNTs were transformed into tetra - diamond NWs under laser irradiation. The dangling bonds of sp³ carbon were passivated by the hydrogen dissociated from methanol. The amorphous - like carbons further adhere to the surface of tetra - diamond NWs (Figure 1b) to act as a passivation layer to preserve the as - grown tetra - diamond NWs. The preservation of the shape of MWCNTs to tetra - diamond NWs might due to the higher absorption of 532nm photons to MWCNTs than methanol solvent. The MWCNTs were

multiphoton excited to high energy and transferred to tetra - diamond NWs and further trapped to room temperature. ” .

The text above tells the story about NWs formation under pulsed-laser irradiation, however, several dark aspects are not clarified, for instance:

- The amorphous structures were generated from methanol or from decomposed NWs?*
- Are the amorphous structures bound to the NWs by covalent bonds?*
- Since methanol solvent does not absorb light at 532 nm (no absorption band for the solvent at this wavelength), what is the relationship between preservation of the shape of MWCNTs to tetra - diamond NWs and higher absorption of 532 nm photons by MWCNTs, especially if absorption at 532 nm by NWs is even higher for the NWs?*
- Could photobleaching of the NWs be expected at long irradiation times, i.e., NWs not completely trapped to room temperature?*

Answer:

The amorphous structures were generated from methanol. ‘The small oxygen *k*-edge can be attributed to the surrounding amorphous structures produced from laser interaction with methanol because methanol is only oxygen source in the reaction system. To further confirm that the oxygen signal is related to the methanol conversion, methanol solvent have been irradiated under the same conditions. Large amount of the same amorphous structures were produced.’ has been added to the right column of page 2.

‘Covalent bonds might be produced between amorphous structures and NWs to passivate the T-carbon NWs during laser irradiation.’ has been added to the right column of page 2.

The absorption spectra of the samples have been shown in Figure S4. The absorption of T-carbon nanowire suspension is much lower than that of MWCNTs suspension at 532 nm. There should be some NWs without amorphous structures if the amorphous structures were produced from the decomposition of MWCNTs or NWs since the suspension is extremely dilute.

‘The detection of hydrogen gas from the reaction further confirmed the decomposition of methanol during laser irradiation since only possible source of hydrogen elements is the methanol solvent.’ has been added to the right column of page 5.

The photobleaching likely happened. The NWs are covered by an amorphous coating produced from methanol conversion. The deposition likely counterbalanced the bleaching to reach a thickness protected the NWs.

Point (16) of Referee 4:

Minor comments:

- The term “psuedo-tpotactic ” can be found several times in the manuscript, however, the term “pseudo-topotactic ” should be used.

- The term “graphene nanoribbons ” should substitute for “grapheme nanoribbons ” in the 1st paragraph of the manuscript.

Answer:

‘psuedo-tpotactic’ has been corrected to be ‘pseudo-topotactic’ all throughout the manuscript and supporting information.

‘grapheme nanoribbons’ has been corrected to be ‘graphene nanoribbons’ in the 1st paragraph.

Additional Revision:

① The title ‘Psuedo-topotactic conversion of carbon nanotubes to tetra-Diamond nanowires in methanol under irradiation of picosecond laser’ has been replaced by

‘Evidence of pseudo-topotactic conversion of carbon nanotubes to T-carbon nanowires in methanol under irradiation of picosecond laser’

② ‘The tetra-diamond structure exhibits a direct band gap of 3.4 eV. The computed electronic band structures and density of states are well consistent with the UV-Vis absorption and photoluminescence of the tetra-diamond NWs.’

has been changed to

‘The change in the entropy from MWCNT to T-carbon reveals the phase transformation to be first order kind. The computed electronic band structures and projected density of states are in good agreement with the UV-Vis absorption and photoluminescence spectrum of the T-carbon NWs.’ in the end of abstract.

③ ‘The HRTEM image is consistent well with its fast Fourier transform (FFT) pattern shown in Figure 1c. The FFT pattern demonstrates an square pattern’ has been replaced by

‘A fast Fourier transform (FFT) pattern calculated from the HRTEM image is shown in Figure 1c. The FFT pattern shows a square pattern’ in the left column of page 2.

④ ‘The reagents in this work only include MWCNTs and methanol. Only carbon, oxygen and hydrogen elements are involved in the products.’

has been replaced by

‘The reagents **in the reaction system** include MWCNTs and methanol **only**. **Hence, only** carbon, oxygen and hydrogen elements are **contained** in the products.’ in the right column of page 2.

⑤ ‘A small oxygen *k*-edge (25±2%) except from carbon *k*-edge was also obtained from the EELS spectrum of NWs.’

has been replaced by

‘A small oxygen *k*-edge **transition** (25±2%) **in addition to** carbon *k*-edge was also **observed** from the EELS spectrum of NWs.’ in the right column of page 2.

⑥ ‘Several absorption peaks between 220-320 nm with three main absorption at 225 nm (5.51 eV), 250 nm (4.96 eV), and 276 nm (4.49 eV) were detected from the tetra-diamond NWs (Figure 4a, black), which were not detected in the methanol solvent after laser ablation (Figure 4a, blue) or reagent suspension (Figure 4a, red).’

has been replaced by

‘**Several absorption peaks were detected from the T-carbon NWs in the range of 220-320 nm (Figure 4a, black). The three main peaks are at 225 nm (5.51 eV), 250 nm (4.96 eV), and 276 nm (4.49 eV). Contrary to the spectrum of T-carbon, there are no peaks presented in the spectra measured from either the methanol solvent after laser irradiation (Figure 4a, blue) or the MWCNT suspension before laser irradiation (Figure 4a, red). The full scale absorption spectra are shown in Figure S4.**’ in the left column of page 4.

Reviewers' comments:

Reviewer #1 (Remarks to the Author):

In the revised version, the authors replied all questions raised by four referees with more additional experimental data added to the text and supporting information. I read all replies and responses to these timely and technically questions, and have an impression that the authors made a nice job in revising their previous presentation. Now, I can conclude that, based on new data and their explanations to ambiguities incurred in the old version, this revised manuscript presents an important work toward understanding the diversity of carbon with solid and consistent evidences by performing various experimental inspections. I am satisfied with their improvements. Therefore, I would like to recommend publication of this manuscript in Nat. Comm.

Reviewer #2 (Remarks to the Author):

The authors have agreed that transformation of MWCNT into T-carbon is a first-order phase transformation induced by picosecond laser irradiation. But, they do not explain how? They believe that it is different from the nanosecond laser one (ref 47), but have no clue about the nature of this first-order phase transformation. This mechanism of T-carbon formation must be addressed in some detail, as it is the most important part of the paper. Other comment, which is related to this critical issue, focuses on topotactic growth. For this to be topotactic growth, MWCNT must provide a template for growth and relate to T-carbon crystallographically. The authors have not explained how is this happening?

Reviewer #3 (Remarks to the Author):

(This reviewer chose to provide confidential remarks to the editor only, but expressed satisfaction that their comments have been adequately addressed.)

Reviewer #4 (Remarks to the Author):

Reviewer comments to manuscript revision Nature Commun

Most of the major concerns raised by this reviewer have been properly addressed in the revised version, therefore, the manuscript could be accepted for publication after minor corrections:

Page 2, 2nd column: "To further confirm that the oxygen signal is related to the methanol conversion, methanol solvent have been irradiated under the same conditions."
... methanol solvent has been irradiated...

Page 3, 2nd column: "The NW was further tilted to have an angle relative to figure 2a of 18° and to Figure 2b of 10°,"... and ... "Only lattice planes with either all even whose sum is a multiple of 4 or all odd indexes were observed in FFT patterns due to the extinction rules for T-carbon, which were demonstrated in supporting information."
... tilted...
... Only lattice planes with either all even, whose sum is a multiple of 4, or all odd indexes were observed in FFT...

Page 4, 1st column: "Contrary to the spectrum of T-carbon, there are no peaks presented in the spectra measured from either the methanol solvent after laser irradiation (Figure 4a, blue) or the MWCNT suspension before laser irradiation (Figure 4a, red). The full scale absorption spectra are shown in Figure S4."
However, a peak is clearly observed for the MWCNT suspension before laser irradiation between

250-300 nm (Figure 4a, red, and Figure S4). Please, modify the text accordingly to this issue. Why are there two baseline lines for the red spectrum of Figure 4a?

Page 5, 1st column: "The mechanism of structure transformation of MWCNTs to T-carbon is believed to be different from that proposed for the transformation of amorphous carbon film to diamond induced by nanosecond laser.⁴⁷"
... induced...

Page 6, 1st column: "Raman spectroscopy was taken in a back-scattering geometry using a single monochromator with a microscope (Reinishaw inVia) equipped with CCD array detector (1024 × 256 pixels, cooled to 570°C) and an edge filter."
... cooled to 570°C ???

Figure S5. Concerning the method used for the determination of the emission quantum yield. It is still quite unclear how the numerical value has been calculated.

"Figure S5. Quantum yield measurement of the T-carbon NWS. The quantum yield is calculated as the ratio of the number of photons emitted (N_{em}) by the number of photons absorbed (N_{abs}). Number of emitted photons is given by the area under the spectrally corrected emission (A_{em}). Number of absorbed photons is given by the difference of areas under the Rayleigh scattering peaks of a reference sample and a sample under study ($N_{abs} = A_{scat\ ref} - A_{scat\ sample}$). $Q = N_{em} / N_{abs} = A_{em} / (A_{scat\ ref} - A_{scat\ sample}) = 5.4 \pm 0.2\%$ ".

What type of reference has been used?

Provided that, page 4, 1st column: "The solution after laser irradiation of MWCNT suspension in methanol was directly used for UV-Vis and PL measurements" and that the solution after laser irradiation still has quite a lot of MWCNTs and amorphous structures able to cause strong light scattering, which is not related to that from T-carbon NWS; is this type of calculation the appropriate one?

The most common method for the determination of emission quantum yields, based on the comparison of the emission bands of the sample and a suitable standard (matching the emission band of the sample, and with the same absorbance at the excitation wavelength in both solutions) for the same emission wavelength window collected, is described in this reference: M. Montalti, A. Credi, L. Prodi and M. T. Gandolfi, Handbook of Photochemistry, 3rd ed.; CRC Press: Boca Raton, FL, 2006; Chapter 10, pp 572–576. I suggest to use this method.

Reply to the Referees

The followings are the details of our replies to the comments and questions raised by the referees. The change has been made to the manuscript and supporting information accordingly.

Referee 1

Recommendation: In the revised version, the authors replied all questions raised by four referees with more additional experimental data added to the text and supporting information. I read all replies and responses to these timely and technically questions, and have an impression that the authors made a nice job in revising their previous presentation. Now, I can conclude that, based on new data and their explanations to ambiguities incurred in the old version, this revised manuscript presents an important work toward understanding the diversity of carbon with solid and consistent evidences by performing various experimental inspections. I am satisfied with their improvements. Therefore, I would like to recommend publication of this manuscript in Nat. Comm.

Referee 2

Recommendation: The authors have agreed that transformation of MWCNT into T-carbon is a first-order phase transformation induced by picosecond laser irradiation. But, they do not explain how? They believe that it is different from the nanosecond laser one (ref 47), but have no clue about the nature of this first-order phase transformation. This mechanism of T-carbon formation must be addressed in some detail, as it is the most important part of the paper. Other comment, which is related to this critical issue, focuses on topotactic growth. For this to be topotactic growth, MWCNT must provide a template for growth and relate to T-carbon crystallographically. The authors have not explained how is this happening?

Answer:

Yes, we have agreed that the transformation of MWCNTs into T-carbon is a first-order transformation induced by picosecond laser irradiation because of the dramatic structural changes involving a great number of bonds breaking and forming during the transformation, and distinct variation of entropy from sp^2 bonds in MWCNTs to sp^3 bonds in T-carbon. However, we hesitated to discuss the mechanism of the transition for reasons including: 1) the scope of this paper is limited to provide evidence for the formation of T-carbon which

have been predicted theoretically, but never observed experimentally; and 2) unlike reference 47, which was involved with the conversion of amorphous carbon film (with some sp^3 bonding) to quenched carbon, i.e. Q-carbon, by nanosecond laser irradiation, our system is more complicated, involved in the conversion of sp^2 multiwalled carbon nanotubes (MWCNTs, no sp^3 bonding to begin with) suspended individually in methanol, where the gas evolution was observed during the conversion. The phase transition by laser melting is a highly kinetically controlled process (D. Bauerle, Laser Processing and Chemistry, Springer Verlag, 2000), therefore it is highly sensitive to the specifications of systems and operation conditions. As authors of Reference 47 pointed out that Basharin et al (Tech. Phys. Lett., 36(2010)559) tried to quench diamond using 1 ms laser melting of HOPG with limited success, but they have observed the formation of nanodiamond by nanosecond laser melting amorphous carbon film deposited on the sapphire (different from previously published mechanism of transforming CNTs to diamonds through carbon onions as intermediate (Carbon 36(1998), 997), and interestingly, their experiment failed when the medium was switched from the amorphous carbon film to HOPG. The authors of Reference 47 attributed the formation of nanodiamonds to a super undercooled state. Clearly, the amorphous carbon film on sapphire substrate combined with nanosecond laser irradiation was a sufficient condition for reaching such a “super undercooled state”, and was not after the medium was switched to HOPG. Since both our medium (liquid suspended MWCNT vs. solid thin film) and laser (picoseconds vs. nanosecond) are distinctively different from the experiment setup in reference 47, the mechanisms of structural transitions in these two works differed significantly as demonstrated by different final structures. One key difference lies in time scales of the energy transfer from laser to MWCNTs and subsequent quenching. Rather an asymmetric quenching environment of reference 47 (air on the surface, carbon in lateral direction and sapphire underneath), we have a symmetric environment—individualized MWCNT surrounded by liquid methanol, the energy adsorption and dissipation is confined to the vicinity of MWCNT with a much small scale [15 nm (D) x 200 nm (L)]. Furthermore, as we have discussed in the manuscript, we have observed H_2 gas evolution and amorphous carbon formation from methanol decomposition. We believed that these are key factors for the formation and stabilization of T-carbon. We are not sure that the word “melting” is still literally true in our case (imaging something more than 4000 K in methanol). Energy state should be a right term of description to discuss samples under laser irradiation. Perhaps,

under our condition, we have reached a cooler “super cooled state”. To determine the corresponding temperature of this state is a very challenge problem, our theoretician colleagues are in the process of developing a comprehensive model for simulating this process. It requires much more work which we believe is beyond the scope of this work. It should note that our attempt for the conversion under the same condition except using nanosecond laser instead of picosecond laser was unsuccessful.

Regarding topotactic transition, the referee is correct that a key feature of a topotactic transition is aspects of crystallographic equivalency before and after reaction. The other feature is shape preservation. In our case, there is no strict crystallographic equivalency relationship between MWCNTs and T-carbon. However, the wire shape of MWCNTs was preserved, nanotubes converted to nanowires with statistically similar diameter distribution. To emphasize the fact, we called the transition pseudo-topotactic transition, we believe is a correct call.

The text of ‘It is well known that pulsed-laser-induced liquid-solid interface reaction is a fast and far from equilibrium process. Instantaneous high temperature and high pressure are produced in the laser-materials interface and then immediately quenched. The metastable carbon allotropes between diamond and graphite can be formed and preserved to be final products. The overall shape of CNT auto templates was preserved to yield NW products instead of nanoparticles by irradiation with picosecond pulsed laser instead of nanosecond one. Much stronger heating effect of nanosecond laser than that of picosecond one results in the products with different morphology. The MWCNTs were multi-photon excited to high energy and transferred to T-carbon NWs and further trapped to room temperature. The mechanism of structure transformation of MWCNTs to T-carbon is believed to be different from that proposed for the transformation of amorphous carbon film to diamond induced by nanosecond laser.’⁴⁷

has been replaced by

‘It is well known that pulsed-laser-induced liquid-solid interface reaction is a fast and far from equilibrium process. Instantaneous high temperature and high pressure are produced in

the laser-materials interface and then immediately quenched. The metastable carbon allotropes can be formed and preserved to be final products. The overall shape of CNT auto templates was preserved to yield NW products by irradiation with picosecond pulsed laser. The transformation mechanism of T-carbon nanowires from MWCNTs remains to be explored. Narayan et al.⁴⁷ observed the conversion of nanodiamond from amorphous carbon film deposited on the sapphire by irradiation with nanosecond laser. The formation of nanodiamonds by a melting mechanism via a super undercooled state was proposed. Instead of a solid thin film of carbon, our starting materials are individualized MWCNTs suspended in methanol, and picosecond laser was used. Rather than an asymmetric quenching environment of thin film transition (air on the surface, carbon in lateral direction and sapphire underneath), we have a symmetric environment—individualized MWCNT surrounded by liquid methanol. The energy adsorption and dissipation are confined to the vicinity of MWCNTs with a much small scale [15 nm (D) x 200 nm (L)]. Furthermore, we have observed gas evolution and amorphous carbon formation from methanol decomposition. Given highly kinetically controlled nature of the reaction (sensitive to any condition changed), the setup in this work would invoke different mechanism of structural transition leading to the formation and stabilization of T-carbon nanowires, in light of the difference lies in time scales of the energy transfer from laser to MWCNTs and subsequent ultrafast quenching. We believed that these were key factors for the formation and stabilization of T-carbon nanowires from MWCNTs. We are not sure that the word “melting” is still literally true in our case (imaging objects with more than 4000K immersed methanol). In this work, the MWCNTs were excited to a very high energy state, perhaps, a cooler “super cooled state”. The shapes of the MWCNTs were remained due to short timescale and less heat effect of picosecond laser, which led to the formation of T-carbon nanowires. To determine the equivalent temperature of this state is a very challenge problem. And it will require much more work which we believe is beyond the scope of this work.’ in the 3rd paragraph of page 5.

The text of ‘The hydrogen dissociated from methanol might play a role of surface sp³ dangling bond passivation. The amorphous-like carbons further adhere to the surface of T-carbon NWs (Figure 1b) to act as a passivation layer.’ in the right column of page 5 is explaining the solvent effects.

Referee 3

Recommendation: (This reviewer chose to provide confidential remarks to the editor only, but expressed satisfaction that their comments have been adequately addressed.)

Referee 4

Recommendation: Reviewer comments to manuscript revision Nature Commun. Most of the major concerns raised by this reviewer have been properly addressed in the revised version, therefore, the manuscript could be accepted for publication after minor corrections:

Answer:

We want to express our thanks again to the referee for his or her time and thoughtful comments.

Point (1) of Referee 4:

Page 2, 2nd column: "To further confirm that the oxygen signal is related to the methanol conversion, methanol solvent have been irradiated under the same conditions."

... methanol solvent has been irradiated....

Answer:

'methanol solvent have been irradiated under the same conditions' has been corrected to be 'methanol solvent **has** been irradiated under the same conditions' in the right column of page 2.

Point (2) of Referee 4:

Page 3, 2nd column: "The NW was further tilted to have an angle relative to figure 2a of 18° and to Figure 2b of 10°," ... and ... "Only lattice planes with either all even whose sum is a multiple of 4 or all odd indexes were observed in FFT patterns due to the extinction rules for T-carbon, which were demonstrated in supporting information."... tilted..... Only lattice planes with either all even, whose sum is a multiple of 4, or all odd indexes were observed in FFT...

Answer:

'The NW was further tilted to have an angle relative to figure 2a of 18° and to Figure 2b of 10°' has been corrected to be 'he NW was further **tilted** to have an angle relative to figure 2a of 18° and to Figure 2b of 10°' in the last paragraph of page 3.

‘Only lattice planes with either all even whose sum is a multiple of 4 or all odd indexes’ has been corrected to be ‘Only lattice planes with either all even, whose sum is a multiple of 4, or all odd indexes’ at the end of page 3.

Point (3) of Referee 4:

Page 4, 1st column: “Contrary to the spectrum of T-carbon, there are no peaks presented in the spectra measured from either the methanol solvent after laser irradiation (Figure 4a, blue) or the MWCNT suspension before laser irradiation (Figure 4a, red). The full scale absorption spectra are shown in Figure S4.”

However, a peak is clearly observed for the MWCNT suspension before laser irradiation between 250-300 nm (Figure 4a, red, and Figure S4). Please, modify the text accordingly to this issue.

Why are there two baseline lines for the red spectrum of Figure 4a?.

Answer:

The text ‘Contrary to the spectrum of T-carbon, there are no peaks presented in the spectra measured from either the methanol solvent after laser irradiation (Figure 4a, blue) or the MWCNT suspension before laser irradiation (Figure 4a, red).’

has been changed to be

‘Contrary to the spectrum of T-carbon, there are no peaks at 225, 250, and 276 nm presented in the spectra measured from either the methanol solvent after laser irradiation (Figure 4a, blue) or the MWCNT suspension before laser irradiation (Figure 4a, red).’ in the right column of page 4.

The extra baseline for the red spectrum of Figure 4a has been deleted. The Figure 4a has been replaced in page 4.

Point (4) of Referee 4:

Page 5, 1st column: “The mechanism of structure transformation of MWCNTs to T-carbon is believed to be different from that proposed for the transformation of amorphous carbon film to diamond induced by nanosecond laser.⁴⁷”

... induced....

Answer:

‘indused’ has been corrected to ‘induced’ in page 5.

.

Point (5) of Referee 4:

Page 6, 1st column: “Raman spectroscopy was taken in a back-scattering geometry using a single monochromator with a microscope (Reinishaw inVia) equipped with CCD array detector (1024 × 256 pixels, cooled to 570°C) and an edge filter.”
... cooled to 570°C ???

Answer:

‘cooled to 570°C’ has been corrected to be ‘cooled to 70°C’ in the experiment section of page 6.

Point (6) of Referee 4:

Figure S5. Concerning the method used for the determination of the emission quantum yield. It is still quite unclear how the numerical value has been calculated. “Figure S5. Quantum yield measurement of the T-carbon NWs. The quantum yield is calculated as the ratio of the number of photons emitted (N_{em}) by the number of photons absorbed (N_{abs}). Number of emitted photons is given by the area under the spectrally corrected emission (A_{em}). Number of absorbed photons is given by the difference of areas under the Rayleigh scattering peaks of a reference sample and a sample under study ($N_{abs} = A_{scat\ ref} - A_{scat\ sample}$). $Q = N_{em} / N_{abs} = A_{em} / (A_{scat\ ref} - A_{scat\ sample}) = 5.4 \pm 0.2\%$ ”.

What type of reference has been used?

Provided that, page 4, 1st column: “The solution after laser irradiation of MWCNT suspension in methanol was directly used for UV-Vis and PL measurements” and that the solution after laser irradiation still has quite a lot of MWCNTs and amorphous structures able to cause strong light scattering, which is not related to that from T-carbon NWs; is this type of calculation the appropriate one?

The most common method for the determination of emission quantum yields, based on the comparison of the emission bands of the sample and a suitable standard (matching the emission band of the sample, and with the same absorbance at the excitation wavelength in both solutions) for the same emission wavelength window collected, is described in this reference: M. Montalti, A. Credi, L. Prodi and M. T. Gandolfi, Handbook of Photochemistry, 3rd ed.; CRC Press: Boca Raton, FL, 2006; Chapter 10, pp 572–576. I suggest to use this method.

Answer:

Thanks for providing the method for the determination of emission quantum yield. The method for the determination of absolute photoluminescence quantum yield used in our experiment is an instrument build in one, which is commonly used in absolute photoluminescence quantum yield measurements (<http://www.nature.com/nphoton/journal/v5/n5/full/nphoton.an.2011.1.html>). The method provided by the referee might be more accurate. However, the accuracy of the absolute photoluminescence quantum yield will not affect our conclusion in this work. We will try to use the method provided by the referee in our future works and try to find the difference between them.

The caption of figure S5 ‘Quantum yield measurement of the T-carbon NWs. The quantum yield is calculated as the ratio of the number of photons emitted (N^{em}) by the number of photons absorbed (N^{abs}). Number of emitted photons is given by the area under the spectrally corrected emission (A^{em}). Number of absorbed photons is given by the difference of areas under the Rayleigh scattering peaks of a reference sample and a sample under study ($N^{abs}=A^{scat}_{ref}-A^{scat}_{sample}$). $Q= N^{em}/ N^{abs}= A^{em}/(A^{scat}_{ref}-A^{scat}_{sample})=5.4\pm 0.2\%$ ’

has been replaced by

‘**Absolute photoluminescence** quantum yield measurement of the T-carbon NWs **using an integrating sphere**. The quantum yield is calculated as the ratio of the number of photons emitted (N^{em}) by the number of photons absorbed (N^{abs}). Number of emitted photons is given by the area under the spectrally corrected emission (A^{em}). Number of absorbed photons is given by the difference of areas under the Rayleigh scattering peaks of a reference sample and a sample under study ($N^{abs}=A^{scat}_{ref}-A^{scat}_{sample}$). $Q= N^{em}/ N^{abs}= A^{em}/(A^{scat}_{ref}-A^{scat}_{sample})=5.4\pm 0.2\%$.^{3, 4}’

The references have also been cited.

Additional revision

Title ‘Evidence of pseudo-topotactic conversion of carbon nanotubes to T-carbon nanowires in methanol under irradiation of picosecond laser’ has been changed to be ‘Pseudo-topotactic conversion of carbon nanotubes to T-carbon nanowires in methanol under irradiation of picosecond laser’ due to the word limit of title.

The reference format has been corrected.

REVIEWERS' COMMENTS:

Reviewer #2 (Remarks to the Author):

My comments have not been addressed fully, but there is an improvement over previous version. I am willing to along with the majority!

Reviewer #4 (Remarks to the Author):

As mentioned in my previous report, the work in its present form meets the criteria of novelty and quality for accepting in Nature Commun.

A last minute point which could be considered is related with Point (6) of referee 4:

- A comment should be added to text in Figure S5 caption stating that, since the sample analyzed is not composed of pure T-carbon NWs, the quantum yield value is only an estimation from the crude product.

Referee 4

As mentioned in my previous report, the work in its present form meets the criteria of novelty and quality for accepting in Nature Commun. A last minute point which could be considered is related with Point (6) of referee 4: A comment should be added to text in Figure S5 caption stating that, since the sample analyzed is not composed of pure T-carbon NWs, the quantum yield value is only an estimation from the crude product.

Answer:

The caption of Supplementary Figure 5:

'Absolute photoluminescence quantum yield measurement of the T-carbon NWs using an integrating sphere.'

has been changed to be

'Absolute photoluminescence quantum yield measurement of the T-carbon NWs, based on the crude products, using an integrating sphere.'